# Non-linear interaction between synchronous generator and GFM controlled wind turbines - Inertial effect enhancement and oscillations mitigation

Chinmayi Wagh[1], Johan  Boukhenfouf[1], Frédéric Colas[1], Luis Rouco[2], and Xavier Guillaud[1]

[1]Univ. Lille, Arts et Metiers Institute of Technology, Centrale Lille, Junia, ULR 2697 - L2EP, F-59000 Lille, France
[2]ETS ICAI-IIT, Universidad Pontificia Comillas, 28015 Madrid, Spain

**Correspondence:** Chinmayi Wagh (chinmayi.wagh@centralelille.fr)

**Abstract.** The integration of grid-forming (GFM) controlled wind turbines into AC grids introduces complex dynamic interactions that significantly influence its behavior on the AC side. This study explores the nonlinear coupling between wind turbines and AC grids and propose strategies for the enhancement of the inertial effect and the mitigation of oscillations which can arise in case of an AC event-such as a grid fault or sudden load change. A simplified synthetic model is developed to elucidate the physical insights of these interactions. The findings reveal that wind turbine dynamics has an impact on the inertial contribution and introduce oscillatory behavior under certain conditions. Advanced control strategies are then proposed. They include the integration of input shaping filters and lead-lag compensation to optimize inertial response and dampen mechanical oscillations. The theoretical analysis, validated through simulation, demonstrates the effectiveness and limitations of these methods in enhancing the AC side behavior without compromising the performance of the mechanical system.

## 1   Introduction

In the twentieth century, the electrical power system has a significant global development and most of the operational principles ensuring the system's stability were established based on the behavior of synchronous machines. The phenomenon of the inertial effect plays a crucial role in stabilizing the grid. A key consequence is the limitation of the Rate of Change of Frequency (RoCoF) during significant imbalance between production and consumption. Moreover, since all the rotating speeds of the machine are linked with the grid frequency, the imbalance between production and consumption is naturally propagated all over the grid through the frequency. Consequently, the frequency droop control allows the primary power of multiple power plants, distributed across a wide geographic area, to be adjusted in order to recover the balance. In summary, the synchronization process of these devices is creating couplings between the grid dynamics and the power sources dynamics which are beneficial to the grid frequency stability.

The emergence of inverter-based resources (IBRs) has brought significant changes in the power system analysis. Indeed, the grid following control strategy, which traditionally governs the operation of these converters, relies on a Phase-Locked Loop (PLL) to synchronize with the grid and operates on fundamentally different principles compared to conventional synchronous machines. The Phase Locked Loop (PLL) allows to estimate the grid angle in real time such as it is possible to exchange

an active and reactive power nearly independently from the grid frequency. The PLL breaks the beneficial coupling for the grid that exists in the synchronous machine. In case of a wind turbine, it is possible to recreate a kind of "artificial" inertial effect. Indeed, the inertial response comes from changing the active power with respect to frequency deviation that utilizes the kinetic energy stored in mechanical part of wind turbine. Various inertial control methods for wind power generation have been discussed for several years (Morren et al., 2006; Wu et al., 2018). In addition to standard natural, step-wise and virtual inertial controls for wind turbine optimized design of MPPT control (Wu et al., 2016) or optimized operation of phase lock loop (Hu et al., 2016) provides an improved inertial response with temporarily enhanced frequency support. However, this operation requires the derivation of the grid frequency, which can introduce delays due to the necessary filtering applied to mitigate measurement noise.

Cardozo et al. (2024) highlights the limits of grid following solutions in fulfilling the fundamental system needs to guarantee stable grid behavior. It is the reason why the TSOs proposed to switch from grid following to grid forming control for the control of the IBRs. Indeed, some of the beneficial properties of the synchronous machine can be recovered and even sometime surpassed with such control. In Europe, ENTSOE (2017) has published a report which specifies the different requirements for the grid forming control. In the UK, a best practice guide has been proposed (ESO, 2023), and in the US, EPRI has also published a paper which defines the performance required for the grid forming control (Ramasubramanian et al., 2023). This non exhaustive list highlights the critical role of grid-forming control in various power grids worldwide.

Different types of grid forming control methodologies were proposed. Droop control, synchronous machine-based control, and other controls are summarized in Rathnayake et al. (2021); Rosso et al. (2021). In the case of a virtual synchronous machine (VSM) control, the virtual inertia and damping coefficient are design parameters, unlike the inherent parameters of a synchronous machine. Due to a stronger damping coefficient, the Grid Forming control (GFM) can damp inter-area oscillation (Baruwa and Fazeli, 2021; Xue et al., 2024). However, in all these studies, the DC bus voltage is considered as constant. When a wind turbine is connected to the DC bus, new types of dynamic couplings can be created. On the AC side, the electromechanical coupling between the grid and the mechanical part of the wind turbine modifies the inertial effect brought to the grid in case of a frequency variation (Xi et al., 2018). On the wind turbine side, the disturbance induced by the inertial effect is propagated to the mechanical structure of wind turbine through variations in the electromechanical torque. Two solutions can be proposed to mitigate this effect. One is by adding a damping effect on the electromechanical torque , but this may lead to a weaker DC bus voltage control (Avazov, 2022; Chen et al., 2022). The other is by increasing the damping coefficient on the grid-forming , but this decreases the effectiveness of the inertial response (Tessaro and de Oliveira, 2019).In Heidary Yazdi et al. (2019), a model emphasizing the coupling between the grid and the wind turbine was proposed for operation in the MPPT zone. It employs a one-mass wind turbine model, which limits the insights into electromechanical couplings.

This paper aims to present a theoretical analysis of the new type of coupling dynamics arising from the interaction between wind turbine grid-forming controls and the power grid. Two significant contributions are made in this paper to improve the comprehension and control of wind turbine interactions with AC grids, which are summarized as follows:

a) Analytical Modeling Innovation: A simplified linearized model is developed that integrates wind turbine and power system dynamics, highlighting the strong coupling caused by grid-forming control—an effect associated with emerg-

ing ENTSO-E grid code requirements. This model offers analytical clarity and control-oriented insights not previously explored.

b) Control Strategy Contribution: Based on the proposed model, a control strategy is designed to reduce the coupling between the wind turbine drivetrain and the grid-forming converter. This strategy mitigates drivetrain oscillations and enhances the system's inertial response, with its effectiveness validated through MATLAB simulations.

A simplified synthetic model is proposed in section 3 which highlights the main couplings and allow for the development of some physical insights on this system. In section 4 it is shown that it is possible to act on the control to mitigate the unwanted dynamic behavior.

## 2 Non-linear model of the system

This section introduces the non-linear model of the test case used for EMT simulation in MATLAB-Simulink to study system dynamics. It consists of a 900MVA synchronous generator and a 900MVA aggregated wind farm under grid forming control both, supplying a load connected through a 200 km transmission line (Shafiu et al., 2006). The wind farm consists of 180 type IV wind turbines of 5MW, with constant wind speed at each turbine. Figure 1 shows the model including the control loops of both machines. The detailed models of each component are presented in the following sections.

### 2.1 Model of the wind turbine

The wind turbine model comprises several elements that describe either the power part or the control. The detailed presentation of these elements can be found in the next subsections.

#### 2.1.1 Mechanical model

The aerodynamic power is converted to mechanical power $P_T$ given by Eq. (1) where $R$ represents the wind turbine blades having radius and $\rho$ the air density.

$$P_T = \frac{1}{2}\rho\pi R^2 c_p\left(\lambda,\beta\right)v_w^3 \tag{1}$$

The power coefficient $c_p(\lambda,\beta)$ is derived from the tip speed ratio $\lambda$ and pitch angle $\beta$ using Eq.(2) (Avazov, 2022).

$$c_p = 0.73(\frac{151}{\lambda_r} - 0.58\beta - 0.002\beta^{2.14} - 13.2)e^{\frac{-18.4}{\lambda_r}} \tag{2}$$

where, $\lambda_r = \frac{1}{\lambda - 0.1\beta} - \frac{0.003}{\beta^3+1}$ and $\lambda = \frac{R\Omega_T}{v_w}$ with $\Omega_T$ as the wind turbine rotational speed and $v_w$ the wind speed. To illustrate, Fig.3a depicts the variation of power coefficient as a function of $\lambda$ with $\beta = 0$.

Therefore, from $P_T$ it is possible to derive the mechanical torque on wind turbine $T_T$:

$$T_T = \frac{P_T}{\Omega_T} \tag{3}$$

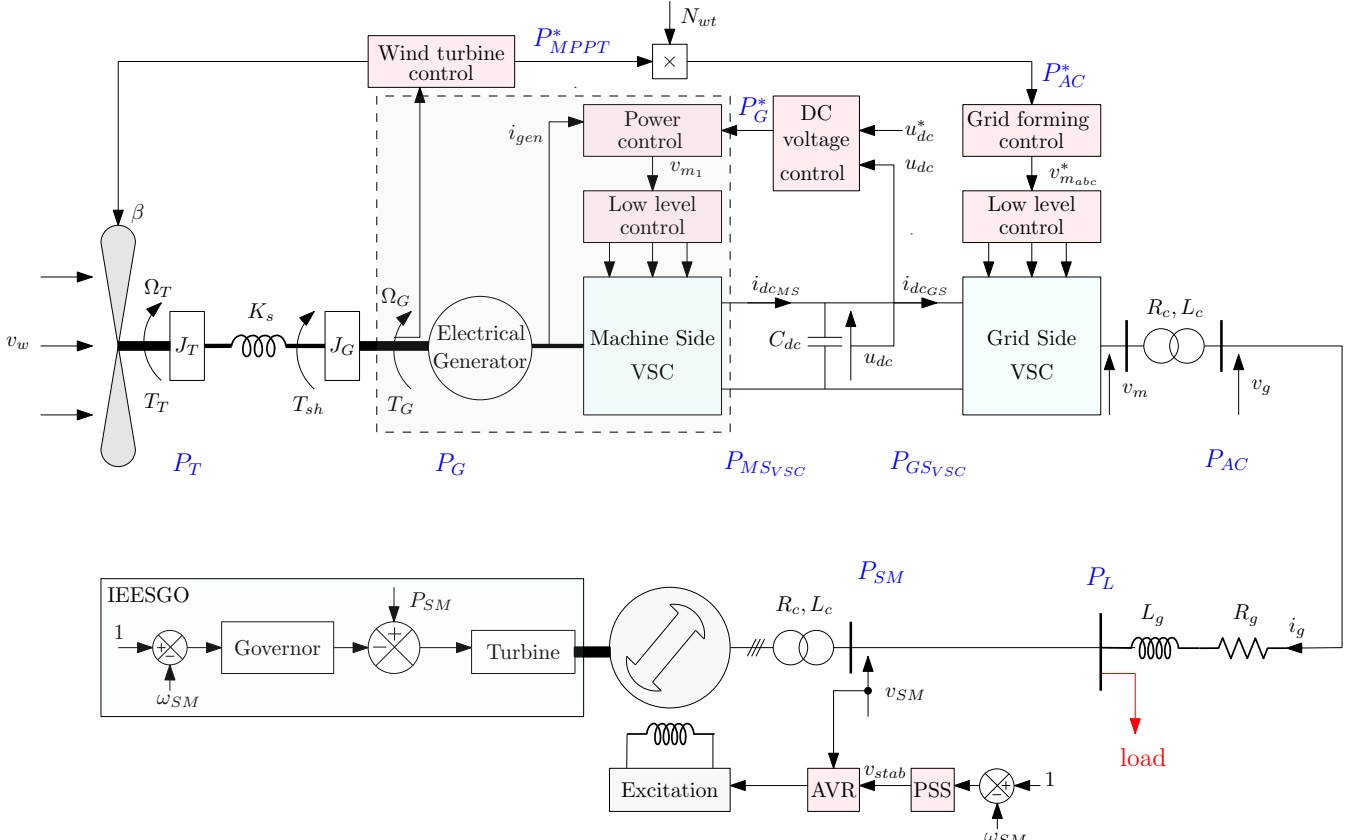

**Figure 1.** Overall system under study

Considering $\Omega_G$ as the rotational speed of the generator and $T_G$ as the torque of the generator, $\Omega_T$ and $\Omega_G$ are derived from the two-mass mechanical model Eq.(4-6) where $J_T$ is the turbine inertia constant, $J_G$ the generator inertia constant, $T_{sh}$ the torque on the shaft, $K_s$ for shaft stiffness and $D_s$ damping coefficient, and $\Delta\omega$ represents the difference in rotational speed between the turbine $\Omega_T$ and the generator $\Omega_G$.

$$\frac{d\Omega_T}{dt} = \frac{1}{J_T}(T_T - T_{sh}) \tag{4}$$

$$\frac{d\Omega_G}{dt} = \frac{1}{J_G}(T_{sh} - T_G) \tag{5}$$

$$\frac{dT_{sh}}{dt} = D_s\frac{d\Delta\omega}{dt} + K_s\Delta\omega \tag{6}$$

Considering the nominal power $P_{nom}$ as the base power $P_b$, and the nominal speed $\Omega_{nom}$ as the base speed $\Omega_b$, the per unit value of the inertia is introduced as $H_T = \frac{0.5 J_T \Omega_b^2}{P_b}$ and $H_G = \frac{0.5 J_G \Omega_b^2}{P_b}$ for the wind turbine and the generator respectively. A per unit coefficient is also introduced for the shaft stifness $K_s^{pu} = \frac{K_s \Omega_b^2}{P_b}$ and the damping coefficient $D_s^{pu} = \frac{D_s \Omega_b^2}{P_b}$. Therefore, Eq.(5) and Eq.(6) can be converted into their per-unit forms:

$$\frac{dT_{sh_{pu}}}{dt} = D_s^{pu} \frac{d\Delta\omega_{pu}}{dt} + K_s^{pu} \Delta\omega_{pu} \tag{7}$$

$$\frac{d\Omega_{T_{pu}}}{dt} = \frac{1}{2H_T}(T_{T_{pu}} - T_{sh_{pu}}) \tag{8}$$

$$\frac{d\Omega_{G_{pu}}}{dt} = \frac{1}{2H_G}(T_{sh_{pu}} - T_{G_{pu}}) \tag{9}$$

For simplicity, the subscript 'pu' for per unit is disregarded throughout the paper. Hence, Fig.2 represents the per-unit mechanical model of wind turbine based on Eq.(7)-Eq. (9).

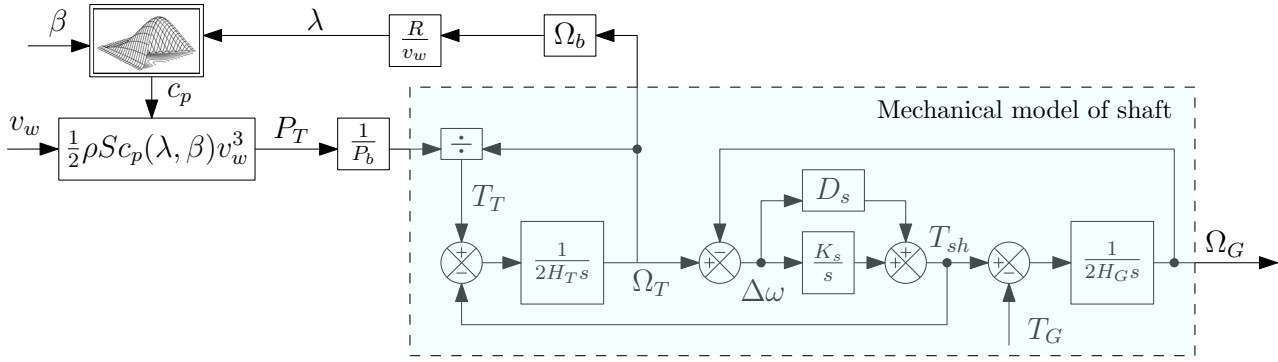

**Figure 2.** 2-mass per unit mechanical model of wind turbine

Additionally, a simplified model can be introduced by neglecting the oscillatory behavior of the turbine. This approach uses a per-unit one-mass mechanical model, where $\Omega_{1mass}$ represents the common rotational speed of both the wind turbine and the generator. Considering their equivalent inertia $H_{1mass} = H_T + H_G$, the shaft is then modeled as:

$$\frac{d\Omega_{1mass}}{dt} = \frac{1}{2H_{1mass}}(T_T - T_G) \tag{10}$$

### 2.1.2 Wind power conversion strategy and optimization

The power curve of the aggregated wind turbine model is assumed same as the power curve of the individual wind turbine. The power curve of the wind turbine has various operating zones represented in Fig.3b in per-unit (Wu et al., 2016; Krpan and Kuzle, 2018) .

In the Maximum Power Point Tracking (MPPT) zone referred as Zone 1, it is possible to define an optimal speed for each wind speed $v_w$ to optimize the power generation. This optimal speed is reached by adjusting the generator torque $T_G$. In this operation mode, the power coefficient and tip-speed ratio have a constant value: $c_{opt}, \lambda_{opt}$ . The maximum power $P_{zone1}$ is given by Eq.(11).

$$P_{zone1} = \frac{1}{2P_b} \rho \pi R^2 c_{p_{opt}} \left(\frac{\Omega_G \Omega_b R}{\lambda_{opt}}\right)^3 \tag{11}$$

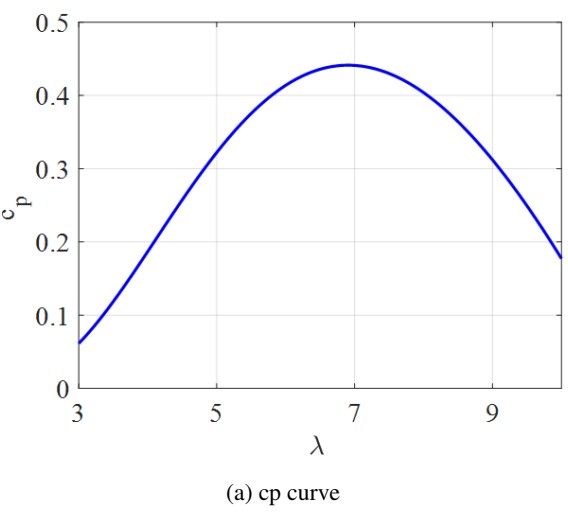

(a) cp curve

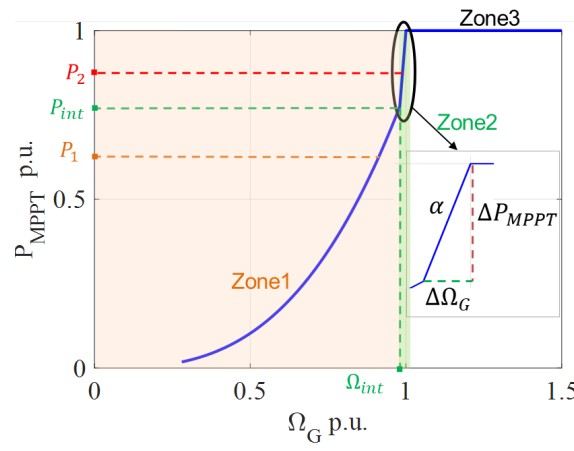

(b) Maximum power point tracking (MPPT) control

**Figure 3.** Wind turbine characteristic curves

After a given wind speed, the MPPT cannot be applied anymore because the nominal rotation speed is reached. Then, the reference power applied aims to stay around the nominal speed, this operation mode is referred as speed limitation: zone 2 between two points $(P_{int}, \Omega_{int})$ and $(1,1)$ with slope $\alpha = \frac{1-P_{int}}{1-\Omega_{int}}$. Then the power $P_{zone2}$ with the slope $\alpha$ is derived as:

$$P_{zone2} = \alpha(\Omega_G - \Omega_{int}) + P_{int} \tag{12}$$

Finally, when the nominal power is reached, the speed cannot be limited by the reference power anymore, the pitch control is actuated to decrease $c_p$ and limit the power generated by the blades to its nominal value. Hence, named as power limitation: zone 3 with power $P_{zone3}$ :

$$P_{zone3} = 1 \tag{13}$$

In summary, the operating point of the wind turbine determines the power output $P^*_{MPPT}$ of the MPPT control given by Eq.(11) or Eq.(12) or Eq.(13) represented in Fig.4.

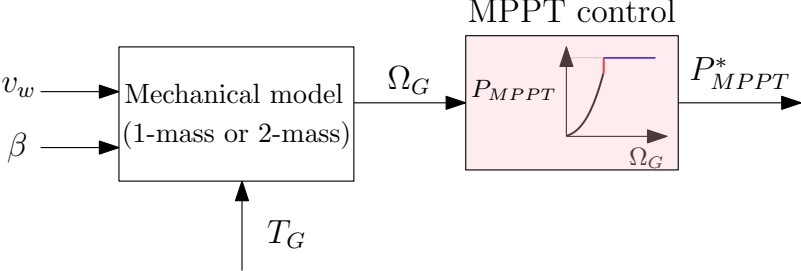

**Figure 4.** Wind turbine MPPT control

### 2.1.3 Model of the machine, machine side VSC and their controls

The generator and machine side VSC are considered lossless; therefore the generator power $P_G$ and power delivered by machine side VSC $P_{MS_{VSC}}$ to DC bus are expressed by Eq.(14) with DC bus voltage $u_{dc}$ and DC current $i_{dc_{MS}}$ injected into the DC bus.

$$T_G \Omega_G = P_G = P_{MS_{VSC}} = u_{dc} i_{dc_{MS}} \tag{14}$$

The electrical power is regulated to a reference value $P_G^*$ through a current loop. Due to the fast dynamics of the control, it can be written that:

$$P_G = P_G^* \tag{15}$$

Hence the DC current $i_{dc_{MS}}$ generated by the machine side VSC is calculated by Eq.(16).

$$i_{dc_{MS}} = \frac{P_G^*}{u_{dc}} \tag{16}$$

And equivalent power $P_{G_{eq}}$ generated by $N_{wt}$ number of wind turbine is:

$$P_{G_{eq}} = P_G N_{wt} \tag{17}$$

### 2.1.4 Grid side VSC model and control

The system is connected to the grid through a VSC. An average model of the VSC is considered, and the control strategy used is Grid Forming. The converter modulates the DC bus voltage in order to generate the three-phase voltage $v_{ma}, v_{mb}, v_{mc}$. The Grid Forming Virtual synchronous machine (VSM) scheme is shown in Fig.5 with virtual inertia $H_{GFM}$ and damping coefficient $K_{d_{GFM}} = \zeta_{GFM} \sqrt{\frac{2(X_g + X_c)}{\omega_b H_{GFM}}}$ designed based on the transmission line impedance $X_g$, transformer impedance $X_c$ and damping factor $\zeta_{GFM} = 0.7$ (Rokrok, 2022). VSM controls the active power $P_{AC}$ to match the reference $P_{AC}^*$ by acting on the modulated voltage reference angle $\theta_m$. The magnitude of the modulated voltage is set to $V^*$. In order to damp the current oscillations, a Transient Virtual Resistor (TVR) $R_v$ modifies the voltage reference $v_{m_{dq0}}^*$ (Lamrani et al., 2023). Then the reference modulated $v_{ma}^*, v_{mb}^*, v_{mc}^*$ are calculated by inverse of Park transformation $P(\theta_m)^{-1}$. Similar to the machine side converter, the grid side converter is assumed lossless with $P_{GS_{MSC}}$ the power exchanged between DC bus and grid side VSC given by:

$$P_{GS_{MSC}} = P_{AC} \tag{18}$$

### 2.1.5 DC bus model and control

The DC bus is modeled using a simple capacitor $C_{dc}$ as:

$$C_{dc} \frac{du_{dc}}{dt} = i_{dc_{MS}} - i_{dc_{GS}} \tag{19}$$

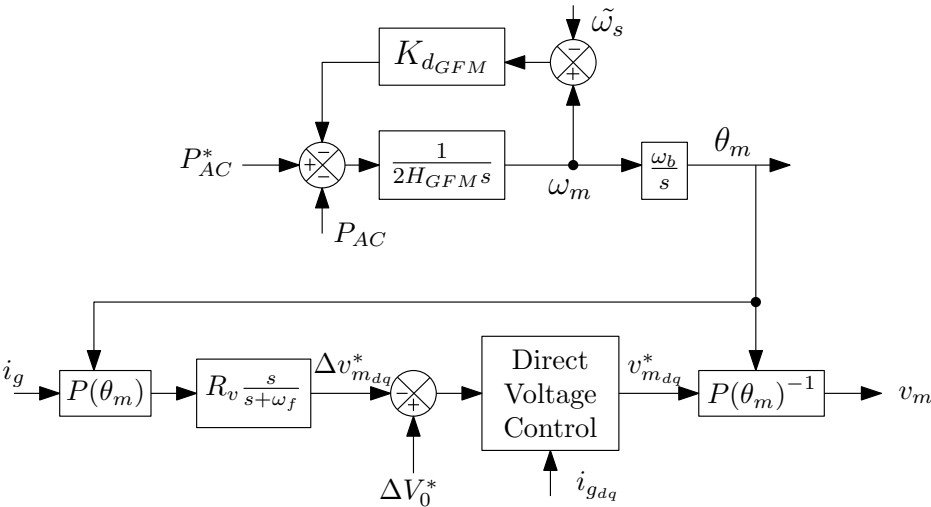

**Figure 5.** Reference model of grid-forming converter

As for the inertia, it is possible to define an $H$ value: $H_{dc} = \frac{C_{dc}V_{dc_b}^2}{2P_b}$ where $V_{dc_b}$ is the base DC voltage equal to the nominal voltage of the DC bus. Hence, the DC bus model in per unit with inertia $H_{dc}$ is expressed as:

$$\frac{du_{dc}^2}{dt} = \frac{1}{H_{dc}}(P_G - P_{AC}) \tag{20}$$

The DC bus control is needed to achieve the stability of the DC bus which depends on the power delivered by the generator side VSC ($P_G$) and the grid side VSC ($P_{AC}$). As demonstrated by Avazov (2022) and Huang et al. (2023), controlling the DC bus voltage through direct action on the generator power is more effective. The control strategy is based on a PI controller.

Hence the output of the DC bus controller is the reference for the generator $P_G^*$ and the reference power of the GFM controller $P_{AC}^*$ for $N_{wt}$ number of wind turbines is derived from Eq.(15) and Eq.(17) as following :

$\quad P_{AC}^* = P_{G_{eq}} = P_{MPPT}^* N_{wt} \tag{21}$

### 2.2 Synchronous machine model

A round rotor type 900MW synchronous machine is modeled using a detailed eighth-order model with $H_{SM}$ as the inertia of synchronous machine and $K_{d_{SM}}$ as the damping coefficient (P.Kundur, 1994). As illustrated in Fig.1 it is equipped with the following:

$\quad$ – a transformer with a resistance $R_c$ and a leakage impedance $L_c$

$\quad$ – an excitation system of type ST1C (exc, 2016) that consist of main voltage regulator with gain $K_a$, a low-pass filter with time constant $\tau_r$ and input voltage $v_{stab}$ from power system stabilizer

- a governor modeled as IEESGO (Pourbeik et al., 2013) with power frequency droop gain $K_{pf}$ and low-pass filters with $\tau_1$ and $\tau_3$ representing the governor dynamics. A three-stage steam turbine is considered with output power $P_{hp}, P_{ip}, P_{lp}$ and a low pass filter with $\tau_4$ to include the dynamics of steam valve demonstrated in Fig.6.

- a powers system stabilizer model PSS1A(exc, 2016) that consist of two lead-lag filters, one wash-out filter $\tau_{p5} = 10s$, one low pass filter $\tau_{p6} = 0.01s$ and PSS gain $K_{PSS} = 0$

## 3 Linear models of the system

In order to use the classical tool for dynamic analysis, such as eigenvalues, a linear model is needed. This involves developing a linearized representation of the wind turbine and its control system. Combining this linearized model with that of the synchronous machine results in the linearized system model shown in Fig.1. By introducing a few additional assumptions, a simplified linearized model can be derived. This simplified model, represented as a block diagram, provides valuable physical insight into the dynamics of the system.

### 3.1 Reference Linear model

The first linear model consists of:

1. Linearized model of wind turbine: It is obtained by linearizing $P_T$ Eq.(1) in two components $\frac{dP_T}{dv_w}$ positive value for all operating points and $\frac{dP_T}{d\Omega_G}$ negative value for all operating points. Even with this simplification, some non-linearities remain in the model. The gain $K_{MPPT}$ represents the power characteristics to generate $\Delta P^*_{MPPT}$:

   (a) In zone1 by linearizing Eq.(11),$\Delta P^*_{MPPT} = \Delta P_{zone1}$ and $K_{MPPT} = \frac{dP_{zone1}}{d\Omega_G}$

   (b) In zone2 by linearizing Eq.(12),$\Delta P^*_{MPPT} = \Delta P_{zone2}$ and $K_{MPPT} = \alpha$

   (c) In zone3 by linearizing Eq.(13),$\Delta P^*_{MPPT} = \Delta P_{zone3}$ and $K_{MPPT} = 0$

2. The grid forming VSC and DC link dynamics model is already linear except the 8th-order model of the synchronous machine which is linearized according to small signal stability analysis (P.Kundur, 1994). The grid is modeled using Kirchhoff's law in the d-q frame for line parameters $R_g, L_g$, and constant impedance load $P_L$

Hereafter for simplicity, the reference linear model is termed as linear model.

### 3.2 Simplified linear model

Using the same linearized model of the wind turbine, the simplified linearized model is developed based on following assumptions:

1. Considering that the energy stored in the capacitor is negligible compared with the energy stored in the inertia's, the dynamics of the DC voltage is neglected. Hence, the power $P_G$ is considered as equal to $P_{AC}$ using Eq.(15), it can be

written as:

$$P_{AC} = P_G^*$$ (22)

2. All the AC system is considered in phasor with active power flow between voltage source $P_{AC}$ and synchronous machine $P_{SM}$ that depends on angle $\delta_m$ and $\delta_{SM}$ corresponding to voltage at grid forming converter $v_m$ and synchronous machine $v_{SM}$ respectively. The quasi-static representation of the grid, as described by Santos Pereira (2020), with load $P_L$ and assuming that voltage angles are small, is as follows:

$$\Delta P_{AC} = \frac{\Delta \delta_m - \Delta \delta_{SM}}{X_{AC} + X_{SM}} - \frac{X_{SM}\Delta P_L}{X_{AC} + X_{SM}}$$ (23)

$$\Delta P_{AC} = \frac{\Delta \delta_{SM} - \Delta \delta_m}{X_{AC} + X_{SM}} + \frac{X_{AC}\Delta P_L}{X_{AC} + X_{SM}}$$ (24)

with $X_{AC} = X_c + X_g$ and $X_{SM} = X_c$ , neglecting the resistance. These equations are used to obtain the network matrix $K_{network}$ as:

$$\begin{bmatrix} \Delta P_{AC} \\ \Delta P_{SM} \end{bmatrix} = \underbrace{\frac{1}{X_{AC} + X_{SM}} \begin{bmatrix} 1 & -1 & -X_{SM} \\ -1 & 1 & X_{AC} \end{bmatrix}}_{K_{network}} \begin{bmatrix} \Delta \delta_m \\ \Delta \delta_{SM} \\ \Delta P_L \end{bmatrix}$$ (25)

3. Simplified model of synchronous machine and governor: A second-order equivalent mechanical model is considered. All other internal dynamics are neglected. The damping coefficient $K_{d_{SM}}$ represents the combined damping effects in the system, which include mainly the effect of the synchronous machine's damper windings, neglected in this simplified model. Due to the non-linearity of the synchronous machine, its value is sensitive to the operating point.

The simplified linearized model of the system is illustrated in Fig.6. This model is only used to explain the different couplings in the system. One of these couplings is inter-area oscillations (in rose area), which primarily occur in large power systems. The other is the effect of using GFM control on the internal behaviour of a single wind turbine. Both phenomena are relevant to the study, but they don't address the same level of power. To merge both models into a single overall model, a strong assumption is made that a full wind farm model is equivalent to a wind turbine model multiplied by the number of wind turbines given in Eq.(17).

The dynamics of linear and simplified linear models are compared with the non-linear model for two operating points $OP_1 = 0.67$ p.u. and $OP_2 = 0.87$ p.u. corresponding to the MPPT zone1 and speed limitation zone2 showed in Fig.3b. Since the operating point of the synchronous machine is modified, the $K_{d_{SM}}$ value has to be adjusted: 30 p.u. for operating point $OP_1$, 10 p.u. for operating point $OP_2$. These values are obtained by the trial and error method by comparing the dynamics

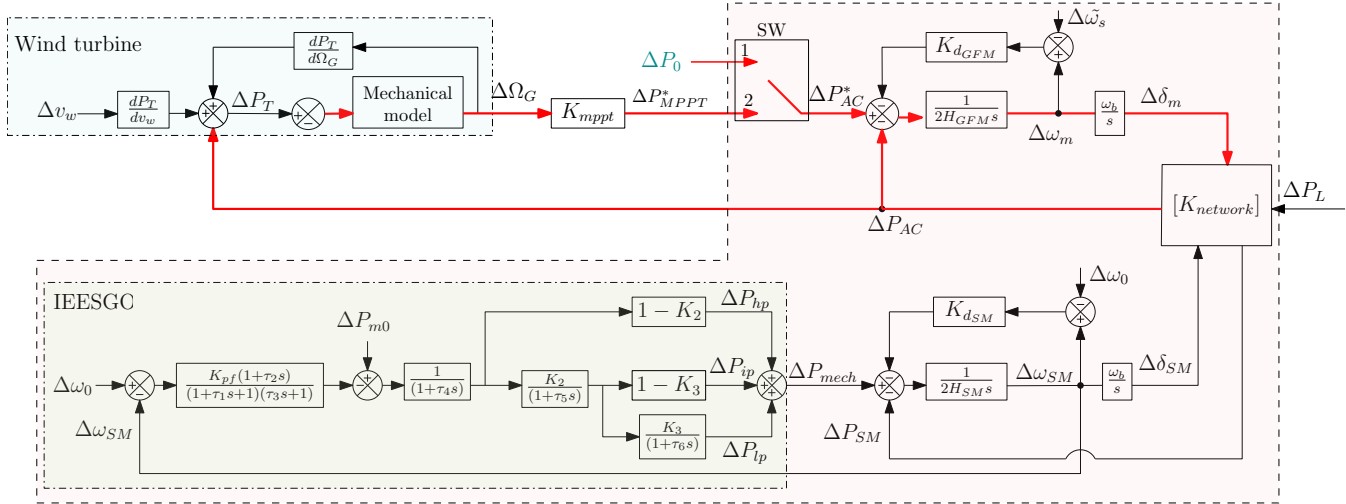

**Figure 6.** Simplified linearized model of the system

with the non-linear model of the synchronous machine. The mechanical model of the wind turbine used is the one-mass model based on Eq.(10). A load step of $\Delta P_L = 0.03$ p.u. is applied at 10 seconds. The resulting variations in system dynamics are shown in Fig. 7 and Fig. 8 for operating points $OP_1$ and $OP_2$, respectively.

Due to the assumption in Eq.(22) used for the simplified linear model, the variation in reference power Fig.7b and DC bus voltage Fig.7d are shown using only the linear model. Figure. 7b and 7e shows that after the load step, the unbalance between power delivered by GFM and generated by the wind turbine slows down the turbine. From Fig.7a and Fig.7b, it is observed that the dynamics of the active power aligns with its reference, thereby confirming the validity of Eq.(22). Whereas Fig.7d illustrates the small variation in DC bus voltage, also true for operating point $OP_2$. Thus, only the variation in AC side power

and wind turbine power is demonstrated in Fig.8 for $OP_2$. Comparing wind turbine dynamics for two operating points, Fig.7c and Fig.8b, reveals a slight variation in $\Delta P_T$ in both linearized models. The first difference comes from the linearization applied to the non-linear model to obtain the linearized model. Then, a second level of simplification is proposed to obtain the so-called "simplified linear" model: the DC link is neglected, and a simple second-order approximation of the synchronous machine is used. The aim of this simplified model is to be sufficiently simple to highlight the main electromechanical couplings

in this complex system, even if some accuracy is lost due to the assumptions made. Further, comparing Fig.7a and Fig.8a it is observed that the inertial response provided by the GFM converter is more for operating point $OP_1$.

The table 1 shows the dominant mode for different wind turbine models. It shows that the simplified model reproduces the dominant frequency modes even though $K_{d_{SM}}$ is different. Therefore, the linear models are validated, and the following sections provide further insight into the effects of variation in the operating zone.

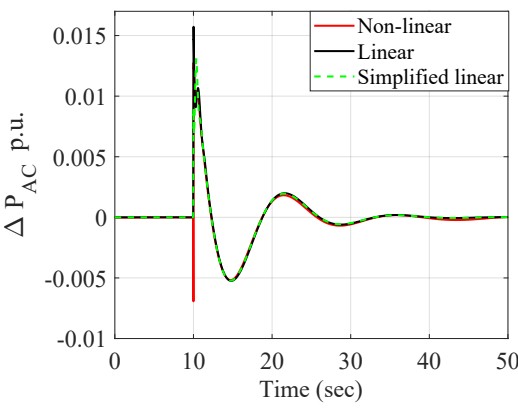

(a) Ac side power $P_{AC}$

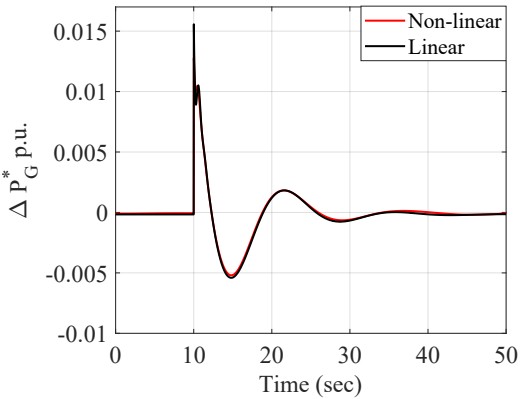

(b) Reference DC power $P_G^*$

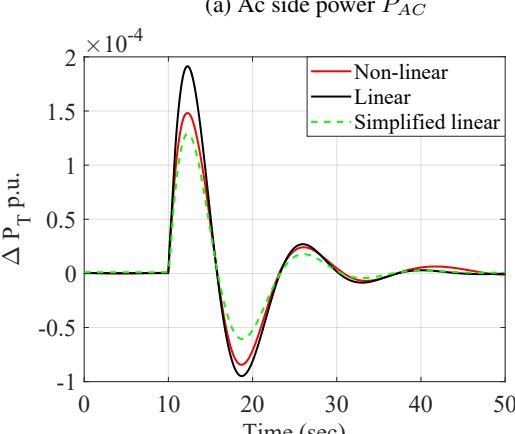

(c) Wind turbine power $P_T$

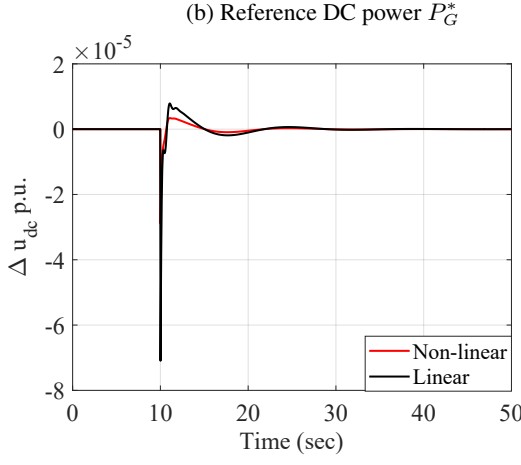

(d) DC voltage $u_{dc}$

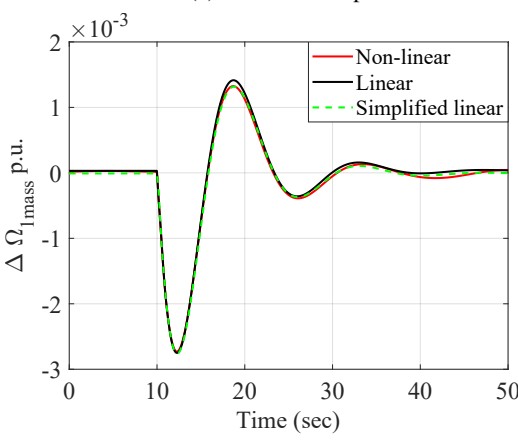

(e) Wind turbine speed $\Omega_{1mass}$

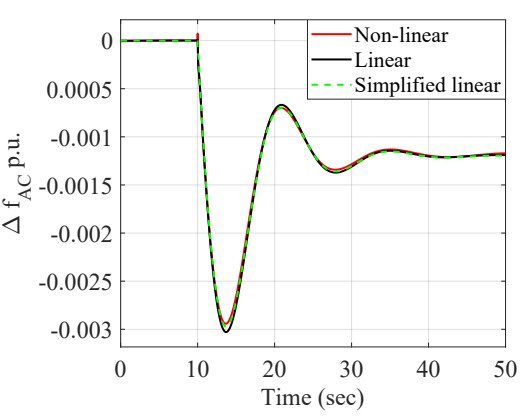

(f) Ac side frequency $f_{AC}$

**Figure 7.** Validation of linear models in zone1

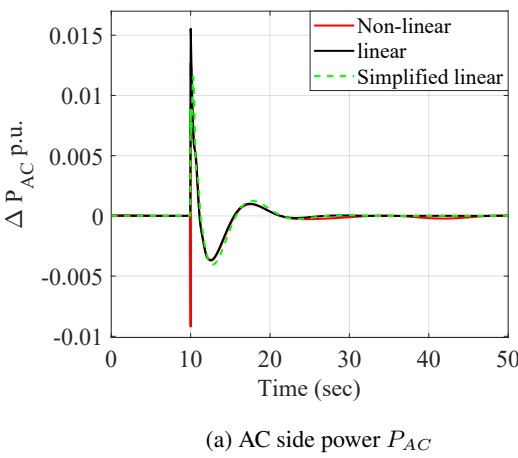

(a) AC side power $P_{AC}$

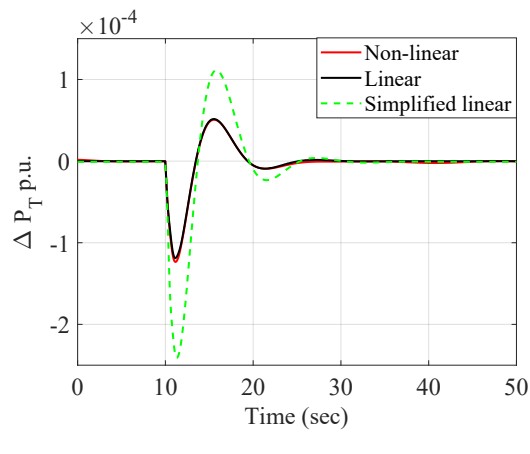

(b) Wind turbine power $P_T$

**Figure 8.** Validation of linear models in zone2

| Model | Constant DC bus-grid forming | 1-mass wind turbine-grid forming | | 2-mass wind turbine-grid forming | |
|---|---|---|---|---|---|
| | | zone1 | zone2 | zone1 | zone2 |
| Linear | -0.22± 0.36j | -0.16±0.44j | -0.3±0.54j | -0.16±0.44j | -0.3±0.54j |
| Simplified linear | -0.23± 0.36j | -0.17±0.44j | -0.28±0.56j | -0.17±0.44j | -0.3±0.55j |

**Table 1.** Comparative analysis of eigenvalues for different systems

## 4 Analysis of the influence of wind turbine on the system dynamics and improvements

### 4.1 Highlighting the main couplings in the system

The simplified linearized model shown in Fig.6 is considered to explain different couplings in the system. In case of a constant DC bus for the Grid Side VSC (Switch SW in position 1 with constant active power reference $\Delta P_0$), this figure reveals a well-known interarea oscillation scheme similar to what can be found with 2 synchronous machines connected via a transmission grid (rose area in Fig.6 ). The main difference is that $H_{GFM}$ and $D_{GFM}$ are controlled variables that can be chosen by the control. Hence, the characteristics of the interarea oscillation modes in terms of frequency and damping can be strongly modified by the choice of these parameters (Baruwa and Fazeli, 2021; Xue et al., 2024).

When SW is in position 2, the simplified linearized model shows that the wind turbine dynamics modify these oscillation modes. Indeed, an additional loop (in red in the system) is added to the system. This loop interacts with the previous modes and modifies the poles and damping as shown in Table 1.

With physical insight from Fig.6 and EMT simulation Fig.9, it can be understood that the wind turbine control tends to counteract the inertial effect brought to the grid by the GFM converter.The control decreases the $P^*_{MPPT}$ signal to recover the

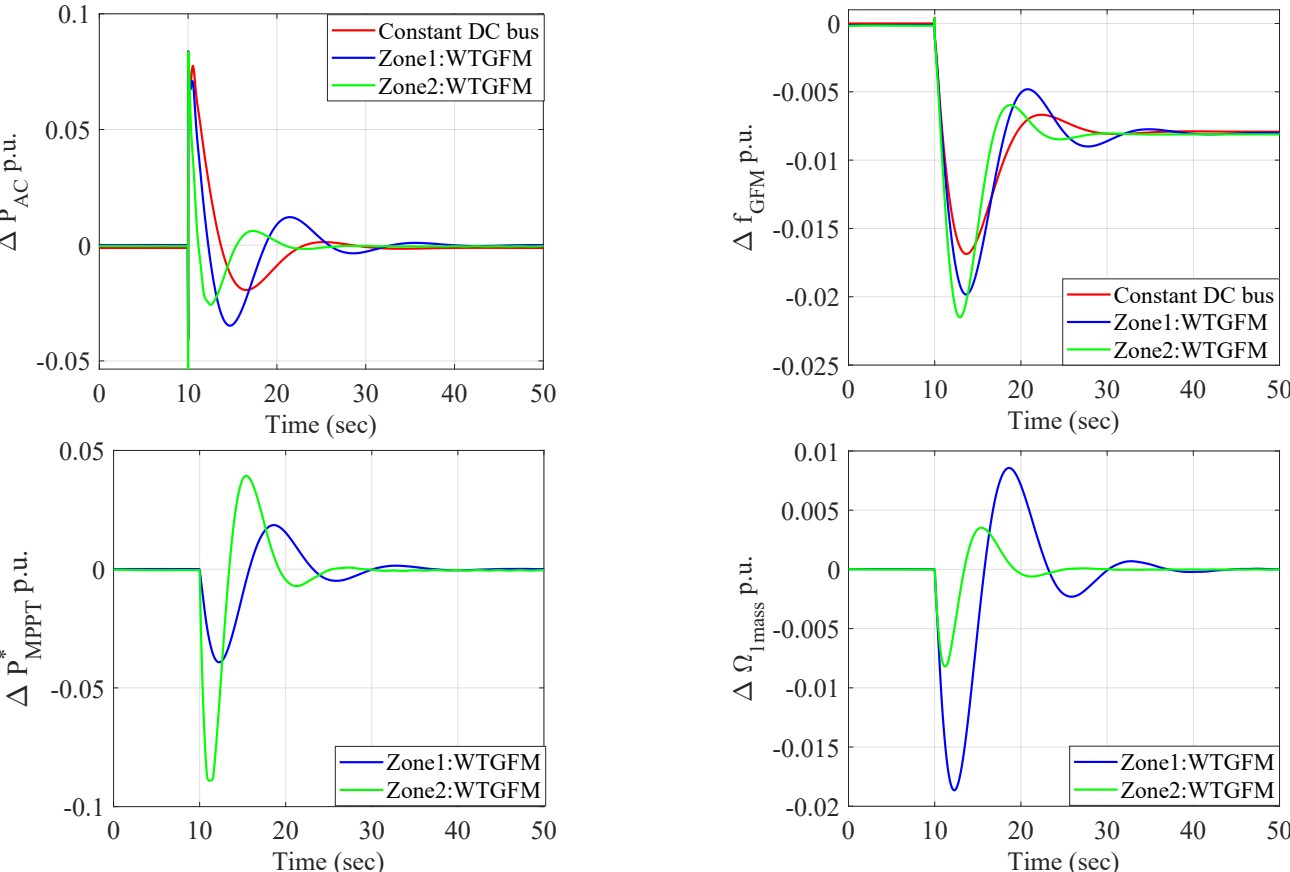

**Figure 9.** Inertial effect and frequency response of system with 1-mass wind turbine for a load step of 0.2 p.u. at 10 sec

optimal rotational speed $\Omega_G$. The effect of this additional loop is to decrease the energy that is exchanged with the grid and to decrease the inertial effect. A possible solution to recover the expected inertial effect will be presented in the next section.

Due to the non-linearity of the loop, the system dynamics vary depending on the operating point. The gain $K_{MPPT}$ increases significantly from Zone 1 to Zone 2, leading to increased system dynamics, as shown in Table 1.

It is also interesting to compare the dynamics of the system by using a one-mass or two-mass model for the drivetrain. The two-mass model offers a more comprehensive understanding of turbine dynamics, particularly in the low-frequency range. In Fig. 10a, a clear decoupling is observed between these eigenvalues and the oscillation modes of the wind turbine itself as the overall grid-side dynamics remain unaffected when employing either a one-mass model or a two-mass model in zone 1. However, in zone 2, where faster dynamics dominate as the equivalent gain $K_{MPPT}$ is higher, the introduction of the two-mass model induces oscillatory behavior shown in Fig.10b.

This section highlighted a new type of coupling due to the connection of a wind turbine to a GFM converter. The resulting undesirable phenomena identified can be mitigated by incorporating dedicated filters into the control loop. Two possible solutions are presented in the following sections.

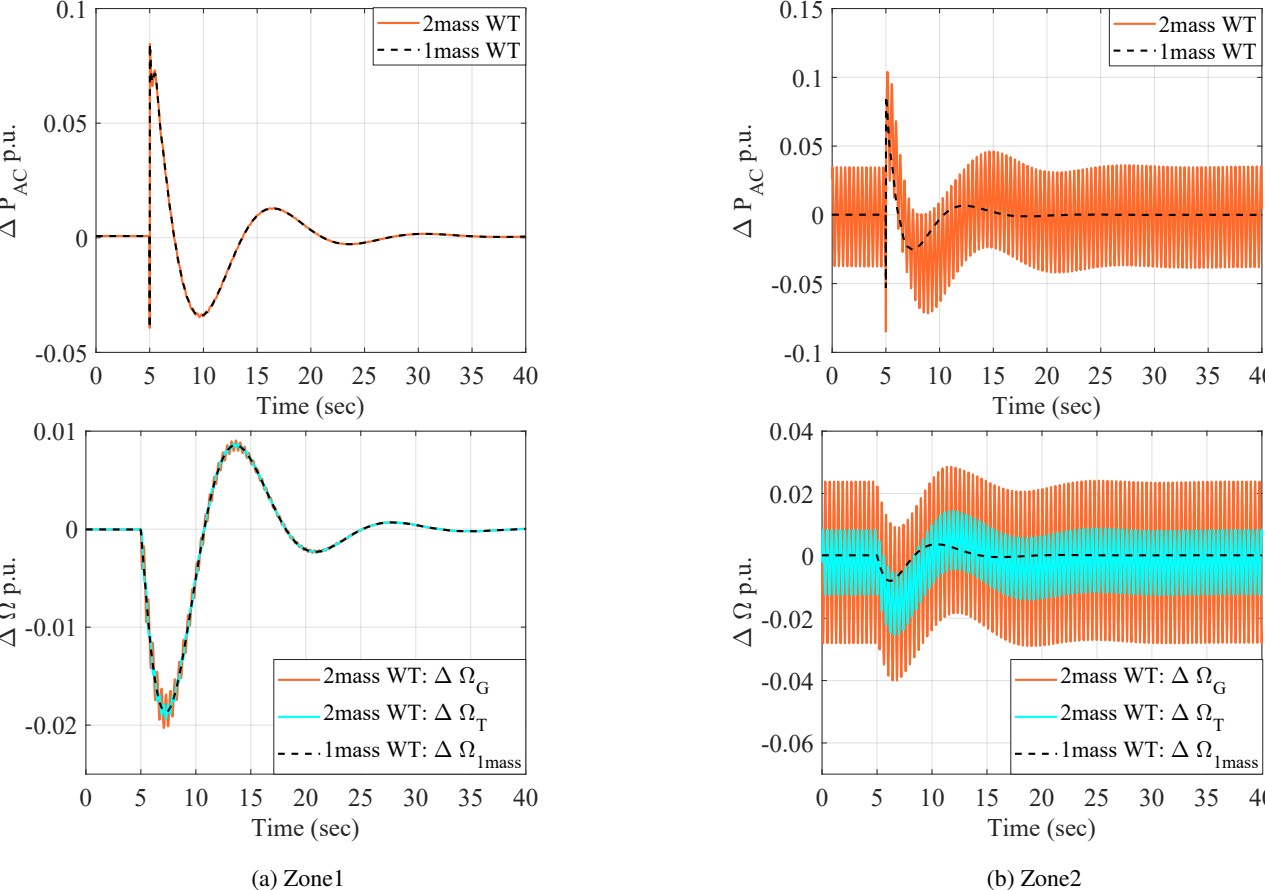

(a) Zone1  (b) Zone2

**Figure 10.** Analysis based on different mechanical system of wind turbine for load step of 0.2 p.u.

## 4.2 Damping the mechanical oscillations using input shaping filter

In Avazov (2022),an input shaping filter was proposed for inclusion in the DC bus voltage control loop. The principle of input shaping is to modify a controlled reference signal by a convolution with a sequence of impulses to get a system response without oscillations. One type of these filters is known as a zero-vibration (ZV) filter allowing to achieve the zero level of residual vibrations (Huey et al., 2008). The design of the filter depends on the number of pulses, their amplitude ($A_k$), and the time of occurrence ($t_k$). A 2-pulse ZV filter as shown in Fig. 11 can be expressed with amplitude $A_1, A_2$ and time $t_1, t_2$ in Laplace domain:

$$IS(s) = A_{1f}e^{-t_1s} + A_{2f}e^{-t_2s} \qquad (26)$$

with the necessary requirements as $0 \leqslant t_1 < t_2$ and $A_{1f} + A_{2f} = 1$.

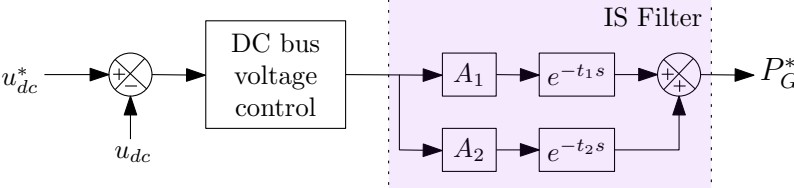

**Figure 11.** DC bus voltage control and input shaping (IS) filter as presented in Avazov (2022)

The resulting finite impulse response filter formed can reduce vibrations in the mechanical system with the drivetrain oscillatory frequency $\omega_{damp}$ corresponding to drivetrain oscillatory mode $s = \sigma + j\omega_{damp}$ as follows:

$$t_1 = 0 \;;\; t_2 = t_1 + \frac{\pi}{\omega_{damp}} \tag{27}$$

Thus, the pulse amplitudes are determined from drivetrain mode are:

$$A_1 = \frac{e^{\frac{\zeta\pi}{\sqrt{1-\zeta^2}}}}{1 + e^{\frac{\zeta\pi}{\sqrt{1-\zeta^2}}}} \;;\; A_2 = 1 - A_1 \tag{28}$$

The drivetrain oscillation frequency of the mechanical system of wind turbine identified from Fig.10b is 15.7 rad/sec. Hence, the ZV filter is designed for $\omega_{damp} = 15.7 \, rad/sec$ using Eq.(27)-Eq.(28), parameters summarized in Table A4. The limitation of this method is to introduce quite a large delay in the control since $t_2$ is quite large and depends on a low frequency mechanical mode. As it has been shown that the energy stored in the capacitor is negligible, the power $P_G$ is nearly equal to $P_{AC}$, another solution is then to place this input shaping filter to generate the reference of the active power shown in Fig.12.

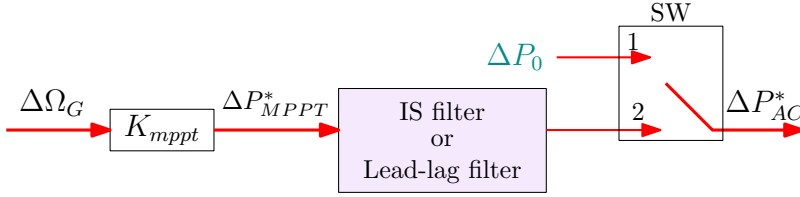

**Figure 12.** Proposed solution: Integration of IS filter in the system

The designed IS filter is tested for variable input wind speed $v_w$ within zone 1 and zone 2 and load step of 0.2 p.u. at 30 sec as shown in Fig. 13. The analysis indicates that drivetrain oscillations, which result from a high gain of $K_{MPPT}$, are effectively damped by the IS filter.

Furthermore, the comparative analysis of the 2-mass model of a wind turbine with IS filter with 1-mass model is presented in Fig.14. This analysis shows that the inertial response of the 2-mass model and the input shaping (IS) control is similar to a 1-mass model. And comparing the wind turbine speed with Fig. 10b proves the effectiveness of IS filter.

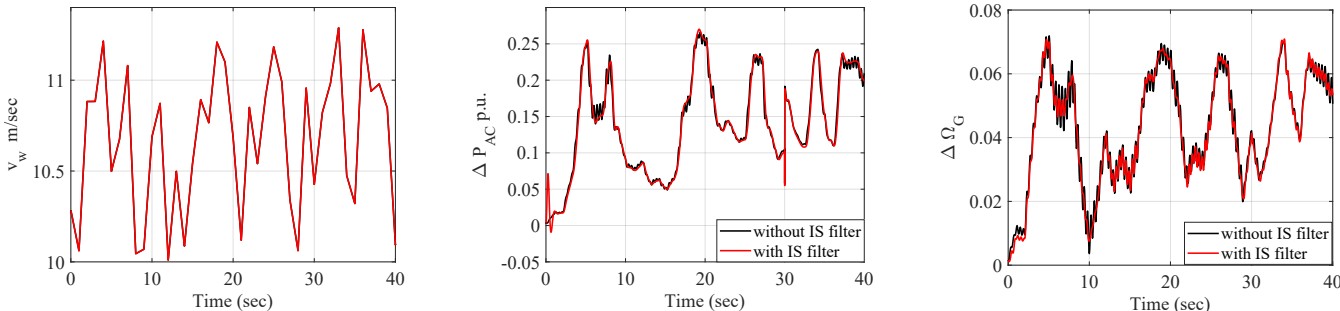

**Figure 13.** Testing of input shaping (IS) filter for variable wind speed $v_w$ and 0.2 p.u. step load at 30 sec

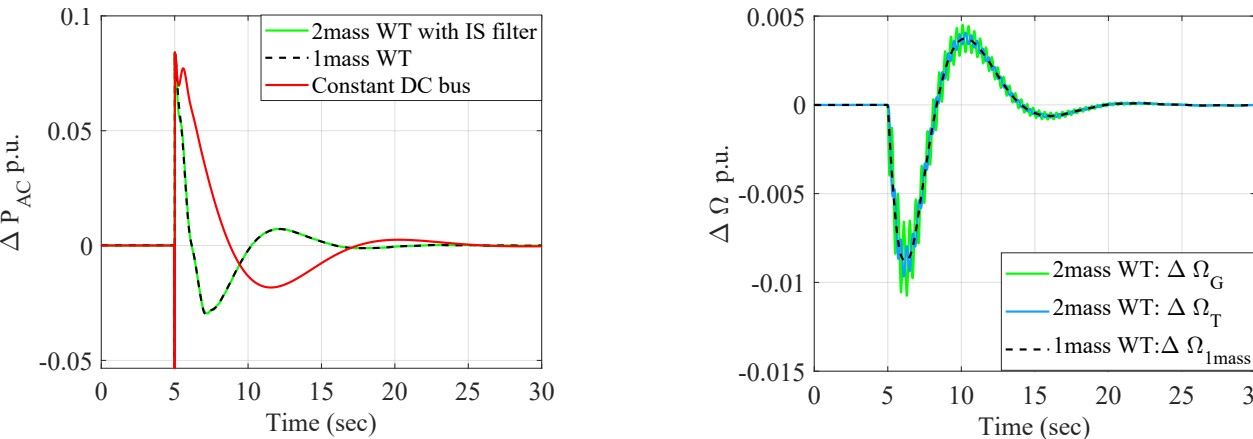

**Figure 14.** Effect of input shaping filter in zone2

### 4.3   Use of a lead lag filter to increase the inertial effect

As described before, the inertial response of a GFM controlled wind turbine is different from GFM controlled VSC with a constant DC bus. To enhance the inertial response of the GFM wind turbine, the MPPT control operation can be deliberately delayed through the implementation of a low-pass filter. This approach serves to smooth the power reference signal, effectively decoupling the GFM inertial effect from the mechanical dynamics of the wind turbine. By introducing this delay, the system mitigates rapid fluctuations in power demand that could otherwise interfere with the turbine's mechanical stability. Additionally, this strategy ensures a more predictable inertial contribution, thereby improving the overall behavior of the wind turbine in response to grid disturbances. This additional low-pass filter is expressed in Eq. (29).

$$\frac{P^*_{MPPT}}{P_{MPPT}} = \frac{1}{1 + t_d s} \tag{29}$$

To correctly tune this filter, the system response has been tested for various lag coefficients ($t_d$) at $OP_1$ for the 0.2 p.u. load step at 10 sec.

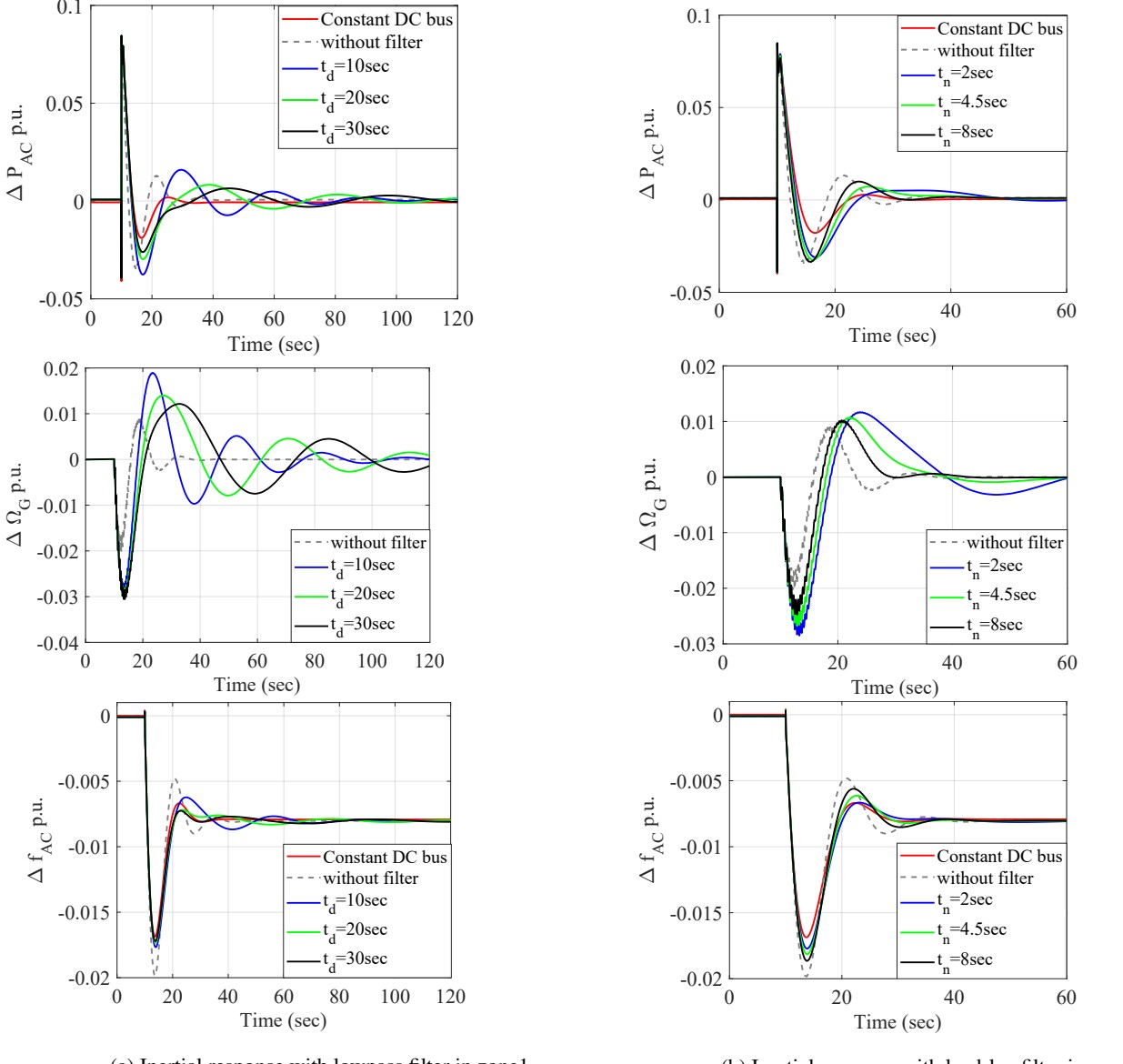

(a) Inertial response with lowpass filter in zone1

(b) Inertial response with lead-lag filter in zone1

**Figure 15.** Inertia enhancement with lowpass and leadlag filter for $OP_1$

The analysis in Fig.15a shows that using a $t_d = 20$ sec low-pass first-order filter provides an improved inertial effect and the frequency response of the AC tends to be the same as with a constant DC bus voltage. It may be possible to increase the filter time constant to 30 seconds, but this would result in slower mechanical modes. In any case, the mechanical counterpart is a larger speed variation of the wind turbine, which can be explained by the fact that the wind turbine provides more inertial effect to the grid. As it can be shown on Fig.15a, adding a filter tends to decrease the damping of the dominant mode. A possible

solution is to add a lead effect Eq.(30) to improve the transient response.

$$\frac{P^*_{MPPT}}{P_{MPPT}} = \frac{1 + t_n s}{1 + t_d s} \tag{30}$$

The system is tested for various lead coefficients ($t_n$) in zone 1 while keeping the delay $t_d = 20$ sec. The choice is a trade off between the improvement of the damping of the mode and the decrease of the inertial effect. The resulting Fig.15b shows that $t_n = 4.5$ sec is an better choice. When the tuning is achieved in zone 1, the effectiveness of the lead-lag filter in zone 2 has to be checked. As previously mentioned, the large increase of the gain $K_{MPPT}$ leads to a more oscillating behavior, as seen in Fig.16, with no possibility of improving this damping. In this figure, it can be observed that the drive train oscillations are well damped by the filter, eliminating the need to add the input shaping filter in series with the lead-lag filter.

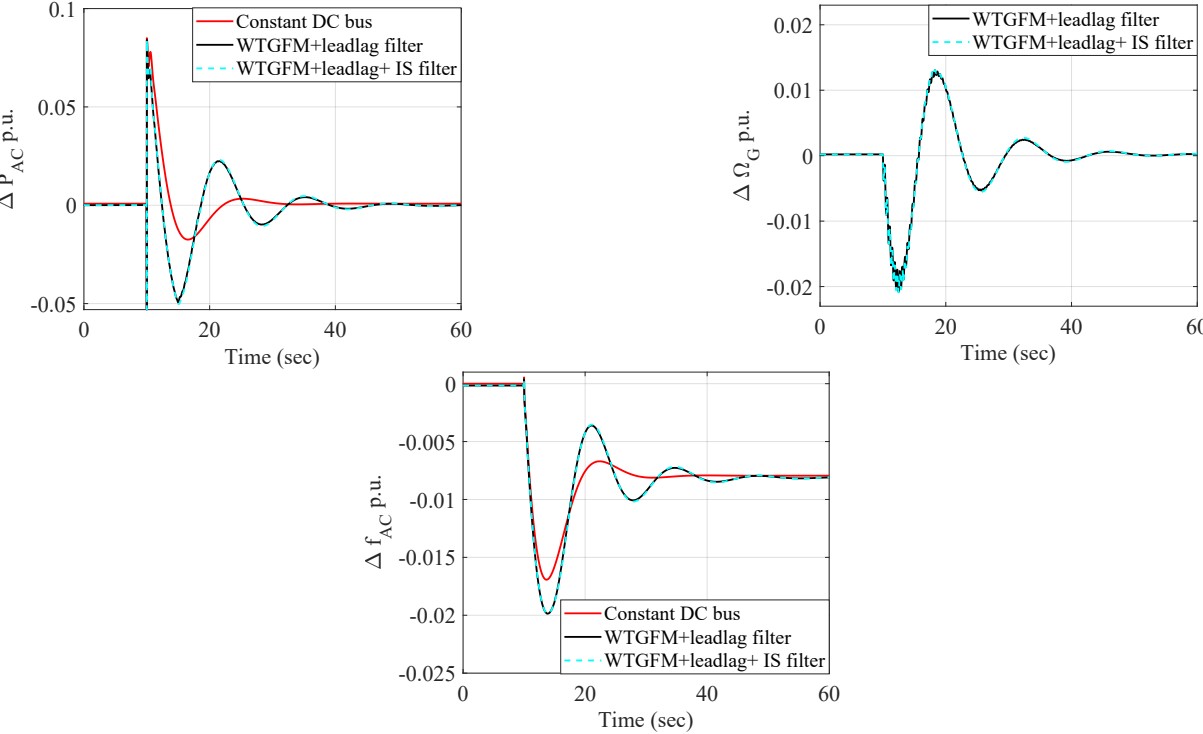

**Figure 16.** Inertial response in zone2 with leadlag filter

### 4.3.1 Robustness assessment under varying system conditions

This section gives an analysis of the performance of the proposed solution for different loading and system conditions. It is presented for both the operating points $OP_1$ and $OP_2$ corresponding to Zone 1 and Zone 2 operation of wind turbine.

For different loading conditions, the load connected near the synchronous machine is increased from 1 p.u. to 1.28 p.u. The Fig.17 shows that the load increase is provided by a synchronous machine, which serves as the swing bus of the system.

Additionally, by comparing $P_{AC}$ it can be seen that the dynamic behavior is less damped for the $OP_2$. The simplified model can help to understand this phenomenon since the gain $K_{MPPT}$ increases significantly from zone 1 to zone 2.

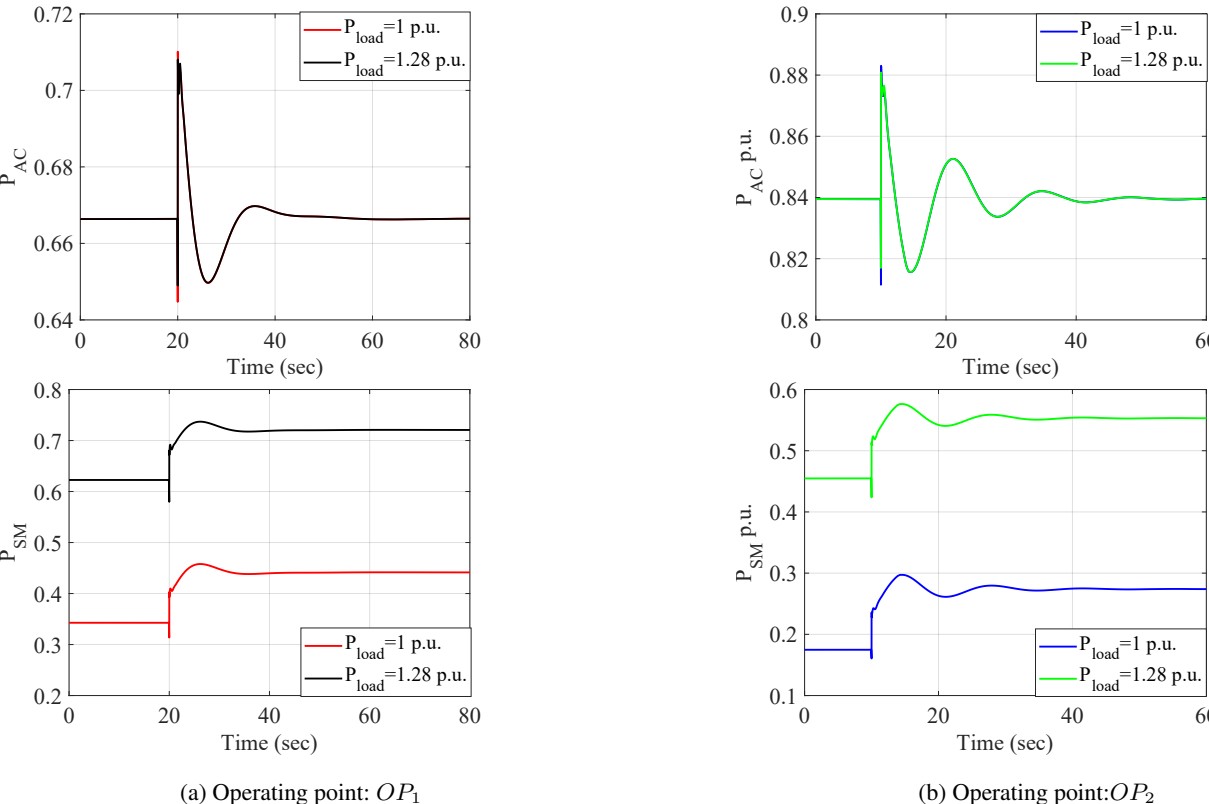

(a) Operating point: $OP_1$                  (b) Operating point: $OP_2$

**Figure 17.** Response for increase in $P_{load}$ for step of 0.1 p.u. at 10 sec

     Secondly, to consider an increased share of wind farms, i.e., a reduced share of synchronous machines investigation with a decrease in the rated capacity of synchronous machines is presented in Fig.18. A noticeable reduction in oscillation damping is evident in the $\Delta\Omega_G$ response for $OP_2$ as $S_{nom}$ decreases. This reduction is attributed to changes in system dynamics resulting

from different system parameters associated with varying $S_{nom}$. Additionally, it is observed that the damping is reduced, and it is challenging to find a good tuning of filter due to high gain $K_{MPPT}$.

     An analysis with different tuning of lead-lag filter is shown in Fig.19, and it is compared with the system without a lead-lag filter and with the original tuning. It is observed that with changes in wind generation, the tuning of the filter can be adjusted, which leads to good damping characteristics of the system. However, this adjustment leads to a reduction in the

inertial response, as seen from $\Delta P_{AC}$. The modification of filter parameters leads to the recovery of the coupling between the wind turbine and the grid-forming converter, as observed from the system response, which becomes closer to the case without a filter after tuning. This is also evident from the closer values of $t_n$ and $t_d$, which results in a weaker lead effect. Therefore, depending on the application, it is a trade-off between the requirement of the initial effect and the damping of the system.

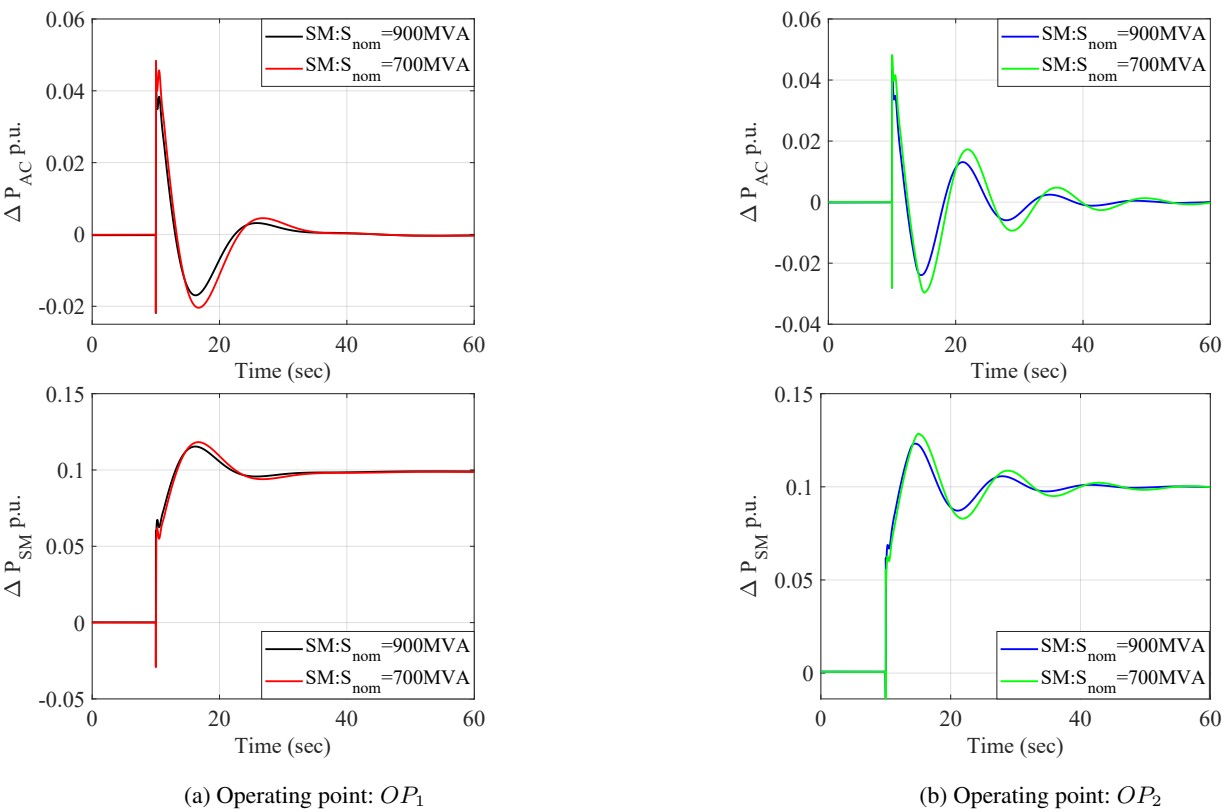

(a) Operating point: $OP_1$

(b) Operating point: $OP_2$

**Figure 18.** Comparative response for reduced capacity of SM for load step 0.1 p.u. at 10 sec

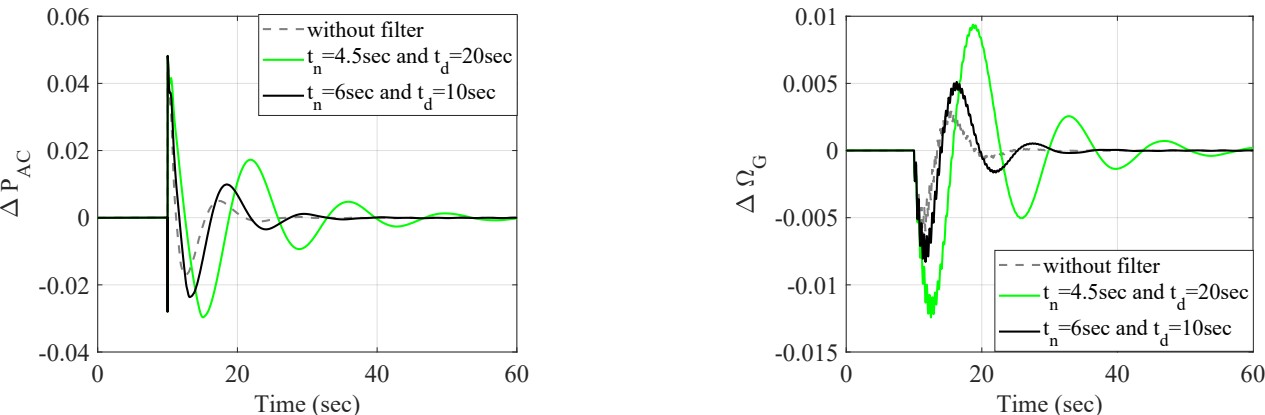

**Figure 19.** $OP_2$: Response of the system for increased share of renewables with modified tuning of lead-lag filter

## 5   Conclusions

The simplified model proposed in this paper highlights the new coupling due to the requirement for grid-forming behavior in wind turbines. This model provides a clear understanding of how the dominant frequency mode is influenced by wind turbine

dynamics and identifies the key parameters that affect this mode. Among these, the gain of the MPPT (Maximum Power Point Tracking) strategy emerges as a critical factor, as its value varies significantly depending on the operational zone.

Additionally, The analysis also underscores the importance of accounting for the internal mechanical dynamics of the wind turbine. Depending on the operating point, unwanted oscillations can appear in the rotational speed of the turbine. To address this, an active damping effect can be introduced to suppress oscillatory mechanical modes by incorporating an input-shaping filter to generate the active power reference. This approach is advantageous compared to previously proposed methods, as it avoids interference with the DC bus control dynamics and has a negligible impact on the active power dynamics of the wind turbine. It has also been further revealed that the internal dynamics of the wind turbine influence the dominant frequency mode. This is due to the interaction between the MPPT control and active power management, as rapid modifications of the power reference reduce the expected inertial effect of the grid-forming converter.

A practical solution to mitigate this issue is the addition of a filter in the control loop. In this case, the filter is a lead-lag type, which serves to decouple the system's dominant mode from the mechanical modes of the wind turbine. As expected, mechanical variations are more pronounced on the wind turbine side, where energy exchange with the grid is higher. This paper has outlined general trends in the dynamics of grid-forming wind turbines connected to an AC grid; however, further work is needed to deepen this analysis. The proposed solution decouples the inertial effect provided by the wind turbine and the grid forming converter, but there is a limitation to use this inertia as it is against the natural behavior of the system. To address this undesired coupling, a more advanced control strategy can be implemented to achieve effective dynamic decoupling. The mechanical model should be improved to ensure that the unmodeled modes do not induce additional interaction with the grid modes. Moreover, it is possible to implement other types of MPPT control and analyse if the same conclusion can be drawn.

## Appendix A: Parameters of the system

| Parameter | Value |
|-----------|-------|
| $S_{nom}$ | 900MVA |
| $f_{nom}$ | 50 Hz |
| $U_{nom}$ | 400kV |
| $R_g$ | 0.0001 p.u./km |
| $L_g$ | 0.001 p.u./km |
| line length | 200 km |

(a) System parameters

| Parameter | Value |
|-----------|-------|
| $V_{dc_{nom}}$ | 640kV |
| $H_{dc}$ | 40msec |
| $\zeta_{dc}$ | 0.7 |
| $T_{r_{dc}}$ | 0.5 sec |

(b) Dc voltage control parameters

**Table A1.** System parameters

| Parameter | Value |
|---|---|
| $H_{GFM}$ | 4.5 sec |
| $\zeta_{GFM}$ | 0.7 |
| $K_{d_{GFM}}$ | 192 p.u. |
| $R_c$ | 0.005 p.u. |
| $L_c$ | 0.15 p.u. |
| $R_v$ | 0.09 |
| $\omega_f$ | 50 rad/sec |

(a) Grid forming converter

| Parameter | Value |
|---|---|
| $P_{nom}$ | 5MW |
| $N_{wt}$ | 180 |
| $\omega_{nom}$ | 1.1905 rad/sec |
| $H_T$ | 1.93 sec |
| $H_G$ | 0.8 sec |
| $K_s^{pu}$ | 280 p.u. |
| $D_s^{pu}$ | 1 |
| $\lambda_{opt}$ | 7 |
| $c_{p_{opt}}$ | 0.44 |
| $R$ | 63 m |
| $\rho$ | 1.22 |

(b) Wind turbine parameters

**Table A2.** Wind turbine-grid forming parameters

| Parameter | Value |
|---|---|
| $H_{SM}$ | 6.5 sec |
| $X_d$ | 1.8 |
| $X_q$ | 1.7 |
| $X_l$ | 0.2 |
| $X_d'$ | 0.3 |
| $X_d''$ | 0.25 |
| $X_q'$ | 0.55 |
| $X_q''$ | 0.25 |
| $T_d'$ | 8 |
| $T_d''$ | 0.03 |
| $T_q'$ | 0.4 |
| $T_q''$ | 0.05 |

(a) Electrical parameters

| Parameter | Value |
|---|---|
| $K_{pf}$ | -25 |
| $K_2$ | 0.7 |
| $K_3$ | 0.4 |
| $\tau_1$ | 0.1 sec |
| $\tau_2$ | 0 sec |
| $\tau_3$ | 0.2 sec |
| $\tau_4$ | 0.05 sec |
| $\tau_5$ | 7.0 sec |
| $\tau_6$ | 0.4 sec |
| $K_a$ | 200 |
| $\tau_r$ | 0.01 |

(b) IEESGO and excitation system parameters

**Table A3.** Synchronous machine parameters

| Parameter | Value |
|-----------|-------|
| $A_1$ | 0.54 |
| $A_2$ | 0.46 |
| $t_1$ | 0 |
| $t_2$ | 0.2 sec |

| Parameter | Value |
|-----------|-------|
| $t_n$ | 4.5 sec |
| $t_d$ | 20 sec |

**Table A4.** Filter parameters

| | |
|---|---|
| Electro-magnetic transient: EMT | Rate of change of frequency :RoCoF |
| Grid-forming converter: GFM | Switch :SW |
| Input shaping: IS | Transient virtual resistor :TVR |
| Inverter-based resources: IBR | Virtual synchronous machine : VSM |
| Maximum power point tracking : MPPT | Voltage source converter : VSC |
| Per unit:pu | Wind turbine -grid forming:WTGFM |
| Phase Lock loop: PLL | Zero vibration : ZV |

**Table A5.** List of acronyms

| | | | |
|---|---|---|---|
| $c_p$ | power coefficient | $P_T$ | wind turbine mechanical power |
| $R$ | radius of wind turbine | $\lambda$ | tip-speed ratio |
| $\beta$ | pitch angle | $\Omega_T$ | rotational speed of wind turbine |
| $\Omega_G$ | rotational speed of the generator | $\Omega_{1mass}$ | rotational speed of 1-mass mechanical model of wind turbine |
| $T_{sh}$ | torque on the shaft of the wind turbine | $T_T$ | mechanical torque on wind turbine |
| $T_G$ | generator torque | $K_s$ | shaft stiffness |
| $D_s$ | shaft damping coefficient | $H_T$ | wind turbine inertia |
| $H_G$ | generator inertia | $H_{1mass}$ | inertia of 1-mass mechanical model of wind turbine |
| $v_w$ | input wind speed | $\rho$ | air density |
| $P_b$ | Base power | $P_{nom}$ | nominal power |
| $\Omega_b$ | base speed | $\Omega_{nom}$ | nominal speed |
| $P_{MPPT}$ | MPPT power | $K_{MPPT}$ | MPPT gain |
| $P_{zone1}$ | MPPT power in zone1 | $P_{zone2}$ | MPPT power in zone2 |
| $P_{zone3}$ | MPPT power in zone3 | $P_{int}$ | initial power point for zone2 |
| $\Omega_{int}$ | initial rotational speed point for zone2 | $\alpha$ | slope of MPPT curve in zone2 |
| $P_{MS_{VSC}}$ | power of machine side VSC | $P_{GS_{VSC}}$ | power of grid side VSC |
| $i_{dc_{MS}}$ | DC current at machine side VSC | $i_{dc_{GS}}$ | DC current at grid side VSC |
| $u_{dc}$ | DC voltage | $H_{dc}$ | DC bus inertia |
| $V_{dc_b}$ | base DC voltage | $C_{dc}$ | DC capacitance |
| $P_G^*$ | reference electrical power | $P_{G_{eq}}$ | equivalent electrical power |
| $N_{wt}$ | number of wind turbines | $H_{GFM}$ | inertia of GFM |
| $K_{d_{GFM}}$ | damping coefficient of GFM | $\zeta_{GFM}$ | damping factor for GFM |
| $R_v$ | transient virtual resistance | $\omega_f$ | filter bandwidth |
| $P_{AC}^*$ | reference power of GFM | $P_{AC}$ | power output at GFM |
| $H_{SM}$ | inertia of SM | $K_{d_{SM}}$ | damping coefficient of SM |
| $R_c$ | transformer resistance | $L_c$ | transformer leakage inductance |
| $R_g$ | transmission line resistance | $L_g$ | transmission line inductance |
| $X_c$ | transformer leakage impedance | $X_g$ | transmission line impedance |
| $P_L$ | constant impedance load | $OP_1$ | wind turbine operating point in zone1 |
| $OP_2$ | wind turbine operating point in zone2 | $P_0$ | constant active power reference |
| $A_1, A_2$ | ZV filter amplitude | $\omega_{damp}$ | drivetrain oscillation frequency |
| $t_d$ | lag time constant | $t_n$ | lead time constant |

**Table A6.** List of nomenclature

*Author contributions.* CW implemented the test case in MATLAB, with JB contributing to the wind turbine-grid forming modeling, LR to the modeling of the synchronous machine, and FC to the implementation of the input-shaping filter. The study was conceptualized by XG and FC. CW and XG prepared the manuscript, with contributions from all co-authors.

*Competing interests.* The authors declare that they have no conflict of interest.

*Acknowledgements.* The authors gratefully acknowledge the Région Hauts-de-France for funding this project.

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
