# Peer review of "Non-linear interaction between synchronous generator and GFM controlled wind turbines - Inertial effect enhancement and oscillations mitigation"

_Wind Energy Science, 2025_

## Author Comment (AC1)

We would like to sincerely thank Referee 1 and Referee 2 for their valuable comments and suggestions. Below, we provide our response and explanations to each comment. We have included a general introduction to provide context for the detailed responses that follow.

We hope that our explanations address the concerns raised and are satisfactory.

Yours faithfully,
The Authors

**General Introduction**

This paper, "Non-linear interaction between synchronous generator and GFM controlled wind turbines - Inertial effect enhancement and oscillations mitigation " for the system shown in Fig.1 presents the following :

[Figure]

Fig. 1: System under study

1. A simplified linear model to provide insights into different couplings in the system of wind turbine-grid forming connected to a synchronous machine.

This model is illustrated in Fig.2.

[Figure]

Fig.2: Simplified linearized model of the system

The pink area represents the coupling of the inter-area oscillation scheme between the grid-forming converter (GFM) and the synchronous machine (SM)

The red line highlights the inertial effect interaction between the wind turbine (WT) and the GFM control. It is the main focus of the paper.

2. A strategy for mitigating wind turbine (WT) drivetrain oscillations and reducing the coupling between the WT and grid-forming converter (GFM) to enhance the inertial response.

**Note:**

- In the following, the results presented are based on the Non-linear simulation model in MATLAB-Simulink. And the results presented are in per-unit (p.u.)
- Operating points:

    OP1 corresponds to Zone 1 with a wind speed of 10 m/s

    OP2 corresponds to Zone 2 with a wind speed of 10.8 m/s

The two operating points OP1 and OP2 are summarized as follows for a base of 900 MVA:

For OP1:

| Device | P |
|--------|-------|
| WT-GFM | 0.669 |
| SM | 0.339 |
| Load | 1 |

For OP2:

| Device | P |
|--------|-------|
| WT-GFM | 0.866 |
| SM | 0.148 |
| Load | 1 |

- The text in red indicates the lines that have been added in the manuscript.

**Explanation to Referee 1 Comments**

**1)    Explain why there might be a rate of change of frequency limit? What if the value is too high?**

In the introduction of this paper, the Rate of Change of Frequency (RoCoF) is discussed in the context of general power systems.  RoCoF is time derivative of power system frequency and relates directly to system inertia thus providing a measure of robustness for electrical grid. A high ROCOF indicates large displacement of voltage angles that endangers secure system operation. Therefore, there is a limit on RoCoF generally within +- 2Hz/s reference given by the European Network of Transmission System Operators ( ENTSOE) [1].

**2)  Looks as if a wind turbine power curve has been used rather than a wind farm power curve. Something that should at least be noted in the pape**r.

Yes, the reviewer is correct. For the simplification of the aggregated model, the same power curve is assumed for each wind turbine. This clarification has been included in section 2.1.2 by adding  the following sentences:

"        The power curve of the aggregated wind turbine model is assumed same as the power curve of the individual wind turbine.   "

 We acknowledge the simplifications made in the model. Our objective was to highlight and illustrate two types of coupling within a unified framework. One of these is inter-area oscillations, which primarily occur in large power systems. The other is the effect of using GFM control on the internal behaviour of a single wind turbine.  Both phenomena are relevant to the study, but they don't address the same level of power. To merge both models in the same overall model, we decided to use this strong assumption that a full wind farm model is equivalent to a wind turbine model times the number of wind turbines. Even if this model is not extremely accurate, the general trends that can be drawn from this aggregated model can be still considered valid.

We have added this explanation in the manuscript in Section 3.2, line 213:

"One of these couplings is inter-area oscillations (in pink area), which primarily occur in large power systems. The other is the effect of using GFM control on the internal behaviour of a single wind turbine.  Both phenomena are relevant to the study, but they don't address the same level of power. To merge both models into a single overall model, a strong assumption is made that a full wind farm model is equivalent to a wind turbine model multiplied by the number of wind turbines given in Eq.(17)"

3) **Section 2 of the paper could be shortened, since there is lots of familiar material here**

It is true that Section 2 recalls familiar material. We tried to write as short as possible, but the modelling presented in Section 2 provides the foundation for the simplified linear model discussed in Section 3. Indeed, all the content of section 2 is used later in the following sections, so it's difficult to determine which content should be removed, as it may be necessary for the explanations in the next sections.

4) **What is the electrical distance between the synchronous machine, load and wind farm, and how might this affect the results, and the parameter selection process?**

The electrical distance for Rg=0.0001 p.u./km and Lg=0.001 p.u./km is 200km between wind turbine-grid forming and synchronous machine. The load is connected near synchronous machine.

This line has been updated in the text in Section 2, line 69, and the related parameters are listed in the appendix:

"It consists of a 900MVA synchronous generator and a 900MVA aggregated wind farm under grid forming control, both supplying a load connected through **a 200 km transmission line** (Shafiu et al., 2006)."

Further analysis of change in line length from 1 km to 200 km has been carried out for a step load of 0.2 p.u. at 5 sec:

[Figure]

[Figure]

Fig.3 : System response for variation in line length

With an increase in line length, both the frequency nadir and active power transfer decrease, while the mechanical oscillation frequency of the wind turbine remains unchanged. Therefore, the parameter selection process does not require modification.

5) **"hit and trial" should read as "trial and error". Can the authors recommend a better approach than "trial and error" to select the parameters, particularly for application to a real system, and where the system configuration, demand levels, etc. may vary over time?**

Thank you for this suggestion. We have replaced "hit and trial" with "trial and error."

In the simplified model, $K_{d_{SM}}$ represents the effect of the dampers of the synchronous machine, whose effects are neglected in this simplified modeling. This is a typical modeling which is used when the study does not need detailed information of the internal behaviour of the synchronous machine

Therefore, $K_{d_{SM}}$ is extracted from a detailed linearized system model through eigenvalue analysis. The damping coefficient obtained from this linear analysis is then validated with nonlinear time-domain simulations. This approach allows the extraction of an equivalent damping coefficient $K_{d_{SM}}$ associated with the synchronous machine modes [2].

To explain this we have modified sentence on line 209 as following:

" The damping coefficient $K_{d_{SM}}$ represents the combined damping effects in the system, which include mainly the effect of the synchronous machine's damper windings, neglected in this simplified model."

6) **Fig. 3b – "region" and "zone" terms are both used interchangeably – best to stick with one term**

The term 'region' has been replaced with 'Zone' throughout the document.

7) **Which tool is used to run the EMT simulations?**

The EMT simulations are performed using MATLAB-Simulink, and this information has been included in Section 2 of the paper.

8) **Fig. 7c and 8b show noticeable differences between the different approaches, but the text doesn't clearly explain why.**

The wind turbine power ($P_T$) in figure 7c and 8b is different for the simplified model because we neglect DC link dynamics and various assumptions made to simplify the synchronous machine, resulting in the observed differences. However, despite these imperfections, the simplified model provides a good representation of stability trends.

Following explanation is added in Section 3.2, line 230:

"Comparing wind turbine dynamics for two operating points, Fig.7c and Fig.8b, reveals a slight variation in $\Delta P_T$ in both linearized models. The first difference comes from the linearization applied to the non-linear model to obtain the linearized model. Then, a second level of simplification is proposed to obtain the so-called "simplified linear" model: the DC link is neglected, and a simple second-order approximation of the synchronous machine is used. The aim of this simplified model is to be sufficiently simple to highlight the main electromechanical couplings in this complex system, even if some accuracy is lost due to the assumptions made."

9) **How should a zero-vibration filter be designed?**

The zero vibration filter is designed based on the drivetrain oscillation frequency of the mechanical system of the wind turbine. We have added more explanation in Section 4.2 line 280 as follows:

" The drivetrain oscillation frequency of the mechanical system of wind turbine identified from Fig.10b is 15.7 rad/sec. Hence, the ZV filter is designed for $\omega_{damp}$ = 15.7rad/sec using Eq.(27)-Eq.(28), parameters summarized in Table A4. "

10) **How are H(GFM) and D(GFM) chosen? Could they become variables during transients?**

$H_{GFM}$ represents the virtual inertia that can be selected based on the requirement. Whereas the damping coefficient $K_{d_{GFM}}$ is based on the given values of $H_{GFM}$ and the damping ratio $\zeta_{GFM}$ that is calculated as follows:

$$K_{d_{GFM}} = \zeta_{GFM} \sqrt{\frac{(X_g + X_c)}{\omega_b H_{GFM}}}$$

Both values of $H_{GFM}$ and $K_{d_{GFM}}$ are fixed and remain constant during the transient response [3]. The following explanation is added in Section 2.1.4, line 140:

"The Grid Forming Virtual synchronous machine (VSM) scheme is shown in Fig.5 with virtual inertia $H_{GFM}$ and damping coefficient $K_{d_{GFM}} = \zeta_{GFM} \sqrt{\frac{(X_g+X_c)}{\omega_b H_{GFM}}}$ designed based on damping factor $\zeta_{GFM}$ = 0.7 Rokrok (2022)."

It is not possible to change these parameters during the transient since the inertial effect is extremely fast.

11) **One reason to provide an inertial (or fast frequency) response is to obtain a revenue stream by satisfying the particular system service definition. The authors may want to quantify the magnitude and duration of the power injection phase, and the duration and depth of the recovery phase after the frequency nadir when evaluating parameter choices. The authors should look at fast frequency reserve definitions in different countries.**

We avoided using the term "fast frequency response" due to its potential ambiguity. For some, it might suggest inertial response—the immediate, physics-based reaction of synchronous machines—while for others, it may imply fast-acting primary control mechanisms like fast-frequency droop.

To avoid this confusion, we preferred to speak about inertial, which is well defined. Moreover, it injects a transient power and it can be used with a minimum impact on the operating point whereas using a fast frequency primary response is supposed to decrease the power at every time to keep a frequency reserve for a while (30 s or more). This is another topic that is not addressed in this paper.

12) **How should the time delay, td, be best chosen, and would it need to vary with system loading conditions, wind turbine (farm) output, share of generation from renewable (converter-based) sources, etc.? It is not clear that the preferred parameter values given in the paper are robust for a much wider range of system conditions and other systems.**

The following analysis shows the robustness of $t_d$ under different loading conditions:

The operating point of WT-GFM is determined by input wind speed $v_w$ . The load increase is provided by a synchronous machine, which serves as the swing bus of the system.

For OP1: an increase of $P_{load}$ to 1.28 p.u. with a step load of 0.1 p.u. at 10 sec

| Device | P |
|--------|-------|
| WT-GFM | 0.669 |
| SM | 0.617 |
| Load | 1.28 |

[Figure]

Fig.4: System response for variation in load for OP1

For OP2 : an increase of $P_{load}$ to 1.28 p.u. with a step load of 0.1 p.u. at 20 sec

| Device | P |
|--------|-------|
| WT-GFM | 0.843 |
| SM | 0.451 |
| Load | 1.28 |

[Figure]

Fig.5: Response for variation in load for OP2

From the results, it can be seen that the dynamic behaviour is less damped for the second operating point. The simplified model can help to understand this phenomenon since the gain KMPPT increases significantly from zone 1 to zone 2.

The response $P_{AC}$ and $P_{SM}$ *of the* Fig.4 and 5 are included in the paper in the new section Section: 4.3.1 Robustness assessment under varying system conditions to explain system response to different loading conditions. Following explanation is added:

"This section gives an analysis of the performance of the proposed solution for different loading and system conditions. It is presented for both the operating points OP1 and OP2 corresponding to Zone 1 and Zone 2 operation of wind turbine.

For different loading conditions, the load connected near the synchronous machine is increased from 1 p.u. to 1.28 p.u. The Fig.17 shows that the load increase is provided by a synchronous machine, which serves as the swing bus of the system. Additionally, by comparing $P_{AC}$ it can be seen that the dynamic behavior is less damped for the OP2. The simplified model can help to understand this phenomenon since the gain $K_{MPPT}$ increases significantly from zone 1 to zone 2. "

It could be possible to propose another set of parameters as shown below:

[Figure]

Fig. 6 OP2 response for an increase in loading condition

From the analysis in Fig. 6, it is observed that another set of filter parameters can be proposed, but the possible modifications are limited. This is because increasing tn (=6sec ) increases the damping as shown by the black curve, but it limits the decoupling effect of the lag filter as observed by the closer response of the filter with tn=6sec with the case without a lead-lag filter.

**13) What would happen to the frequency response if the wind share was increased, larger load disturbance, weaker grid connection, fewer synchronous machines online, etc. ? What would be the implications for controller configuration and parameter selection?**

We acknowledge that this paper does not address large disturbance scenarios involving current limitations. This remains an important area of investigation and is planned for future research.

To consider increased share of wind farm and reduced share of synchronous machine we analyse the system with 20 % decrease in the rated capacity of synchronous machine as follows:

For OP1: a step load of 0.1 p.u. at 10 sec

[Figure]

Fig.7: Response of the system for increased share of renewables at OP1

The analysis in Fig.7 reveals that decreasing the $S_{nom}$ of the synchronous machine leads to a reduced inertial response for the same step load, causing a higher frequency nadir and a larger variation in $\Omega_G$. This was expected since the wind turbine contribution to the overall inertial effect is a little bit smaller than with a synchronous machine. However, the overall dynamics is not strongly changed.

For OP2 : step load of 0.1 p.u. at 10sec

[Figure]

Fig. 8: Response of the system for increased share of renewables at OP2

 A noticeable reduction in oscillation damping is evident in the $\Delta\Omega_G$ response for OP2 (Fig. 8) as $S_{nom}$ decreases.  This reduction is attributed to changes in system dynamics resulting from different system parameters associated with varying $S_{nom}$ . Additionally, it is observed that the damping is reduced, and it is challenging to find a good tuning of filter due to high gain $K_{MPPT}$ .

An analysis with different tuning of lead-lag filter is shown below, and it is compared with the system without a lead-lag filter and with the original tuning :

[Figure]

[Figure]

Fig. 9 : OP2: Response of the system for increased share of renewables with modified tuning of lead-lag filter

From the analysis in Fig. 9, it is observed that with changes in wind generation, the tuning of the filter can be adjusted, which leads to good damping characteristics of the system. However, this adjustment leads to a reduction in the inertial response, as seen from $\Delta P_{AC}$. The modification of filter parameters leads to the recovery of the coupling between the wind turbine and the grid-forming converter, as observed from the system response, which becomes closer to the case without a filter after tuning. This is also evident from the closer values of tn and td, which result in a weaker lead effect.

Therefore, depending on the application, it is a trade-off between the requirement of the initial effect and the damping of the system.

The response $\Delta P_{AC}$ $and$ $\Delta P_{SM}$ $from$ Fig.7 and 8 and response $\Delta P_{AC}$ and $\Delta \Omega_G$ $from$ $Fig$.9 are included in the paper in the new section Section: 4.3.1 Robustness assessment under varying system conditions to explain system response for different share of renewable energy sources. The following lines are included in the paper:

" Secondly, to consider an increased share of wind farms, i.e., a reduced share of synchronous machines investigation with a decrease in the rated capacity of synchronous machines is presented in Fig.18.

A noticeable reduction in oscillation damping is evident in the $\Delta \Omega_G$ response for OP2 as $S_{nom}$ decreases. This reduction is attributed to changes in system dynamics resulting from different system parameters associated with varying $S_{nom}$. Additionally, it is observed that the damping is reduced, and it is challenging to find a good tuning of the filter due to high gain $K_{MPPT}$.

An analysis with different tuning of the lead-lag filter is shown in Fig.19, and it is compared with the system without a lead-lag filter and with the original tuning. It is

observed that with changes in wind generation, the tuning of the filter can be adjusted, which leads to good damping characteristics of the system. However, this adjustment leads to a reduction in the inertial response, as seen from $\Delta P_{AC}$. The modification of filter parameters leads to the recovery of the coupling between the wind turbine and the grid-forming converter, as observed from the system response, which becomes closer to the case without a filter after tuning.  This is also evident from the closer values of $t_n$ and $t_d$ which results in a weaker lead effect. Therefore, depending on the application, it is a trade-off between the requirement of the initial effect and the damping of the system."

**14) Various time delays are considered, but how robust are the choices against changes in system conditions, demand levels, generator locations, grid strength, renewable share of demand, …?**

Based on the analysis in the previous sections—Fig. 4 and Fig.5 under varying loading conditions, and Fig. 7 and Fig.8 reflecting changes in the synchronous machine's $S_{nom}$—it can be concluded that the selected parameters for the lead-lag filter provide sufficient damping and inertial response owing to changes in system conditions.

As described before, we have added a small section about the analysis in our manuscript:

"Section: 4.3.1 Robustness assessment under varying system conditions"

**15) Be careful when using the term "optimise". Are the presented results "optimal", or just "better"? It looks as if they are simply "better"**

We have reconsidered the use of the term 'optimise' and revised the wording accordingly. About Figure 14b in the paper, $t_n$ = 4.5 s is identified as providing a better overall response compared to other values considered. This is based on a comparative evaluation of the system's dynamic behaviour for different $t_n$ values. The 4.5 s case offers the most balanced trade-off between enhanced inertia effect and overall system performance, making it a more effective choice among the options evaluated.

**16) Figure 15 – are the authors stating that an input shaping filter is not applied when the turbine is operating in zone 2? What happens if the input wind speed is such that the turbine varies continuously between zone 1 and zone 2?**

In Figure 15 of the paper, we present the results with and without the input shaping (IS) filter. The comparison shows that the response with the IS filter combined with

the lead-lag filter is similar to the response using only the lead-lag filter . This suggests that the IS filter provides additional flexibility when used alongside the lead-lag filter in the system.

Following is the analysis for random wind speed between zone 1 and zone 2 and a load step of 0.2 p.u. at 30 sec:

[Figure]

Fig.10: System response to variation in wind speed around OP2

The results in Fig.10 indicate that the IS filter effectively dampens the drivetrain oscillations, leading to an improved system response.  These results are added in Section 4.2.

**17) Appendix A – 50 Hz and 60 Hz are both mentioned – what is happening here?**

The mention of 60 Hz was a mistake. 50 Hz refers to the system base frequency and $\omega_f = 50$ rad/sec is the bandwidth for virtual transient resistance in grid forming converter.

**18) English is rather clunky throughout the paper. It would be helpful to tidy up the English, and to expand upon the explanations of what the results mean, and why they are considered significant.**

We have revised the language for clarity, marked in blue colour, and expanded on the explanations of the results, highlighting their meaning and significance.

**19) Define all acronyms**

All the acronyms are listed in appendix.

**20) Kirchhoff is spelt like so**

Thank you for bringing that to our attention. We have corrected the spelling of Kirchhoff.

**21) From a presentational perspective, the results figures should be placed much closer to the relevant text. For example, at the moment, the figures associated with Section 3 actually appear in Section 4. Consequently, the second half of the paper is not easy to follow, with constant switching between "text" pages and much later "figure" pages.**

We have rearranged the figures for clarity.

**22) Harvard referencing uses 2 formats, e.g. Wagh et al. (2025) or (Wagh et al., 2025). The first format should be used if the reference forms an "active" part of the sentence, and the second format should be used if the reference doesn't form an active part of the sentence. There are many places in the paper where the reference formatting is incorrect.**

Thank you for the clarification. We've made the necessary adjustments to the formatting.

In addition to above following lines are added in the conclusion Section 5, line 349:

"This paper has outlined general trends in the dynamics of grid-forming wind turbines connected to an AC grid; however, further work is needed to deepen this analysis. The proposed solution decouples the inertial effect provided by the wind turbine and the grid forming converter, but there is a limitation to use this inertia as it is against the natural behavior of the system. To address this undesired coupling, a more advanced control strategy can be implemented to achieve effective dynamic decoupling."

**Explanation to Referee 2 Comments**

1) **I would appreciate it if the authors make more clear what they see as the innovative contribution(s) of the paper.**

Thank you for your valuable comment. We appreciate the opportunity to clarify the main innovative contributions of our work. The key novelties of the paper are twofold :

a) **Analytical Modeling Innovation:** We develop a simplified yet insightful **linearized model** (Fig.2) of a wind turbine–grid-forming converter–synchronous machine system. This model merges a power system level modelling including **inter-area oscillations** and **inertial interactions** and the wind turbine level modelling and highlitghts in a single model the coupling between both. This new type of coupling is associated with the use of grid-forming control for wind turbines, which will be required by the new grid codes currently being developed by ENTSO-E. (European Network of Transmission System Operators  https://www.entsoe.eu/) .  And the consequence is an increase stress on the mechanical part. Due the non linearity of the wind turbine model, the dynamics are strongly modified depending on the operating point. To the best of the authors' knowledge, this analysis has not been conducted before with such depth. To sum up, each part of the model, wind turbine, power system, is quite simple the originality of the proposed model is to highlight the strong coupling between the two levels. Unlike existing nonlinear or simulation-based approaches, this simplified formulation offers analytical transparency and facilitates control-oriented understanding of the system behaviour.

b) **Control Strategy Contribution**: Thanks to the proposed model, it has been possible to highlight clearly the main coupling and submitted to contradictory requirements between the contribution to electrical power system needs, inertial effect and management of the dynamics of the wind turbine,  propose a control strategy that effectively reduces the coupling between the WT drivetrain and the GFM control loop. This helps to mitigate drivetrain oscillations and enhances the contribution of the GFM to system inertia. The effectiveness of this strategy is demonstrated through MATLAB simulations. This is the second novelty of the paper.  A more in depth discussion about the way to derive the control is proposed in the comment 8)

This is summarized in the introduction on line 55 :

"Two significant contributions are made in this paper to improve the comprehension and control of wind turbine interactions with AC grids, which are summarized as follows:

a) Analytical Modeling Innovation:
A simplified linearized model is developed that integrates wind turbine and power system dynamics, highlighting the strong coupling caused by grid-forming control—an effect associated with emerging ENTSO-E grid code requirements. This model offers analytical clarity and control-oriented insights not previously explored.

b) Control Strategy Contribution:
Based on the proposed model, a control strategy is designed to reduce the coupling between the wind turbine drivetrain and the grid-forming converter. This strategy mitigates drivetrain oscillations and enhances the system's inertial response, with its effectiveness validated through MATLAB simulations."

2) **Some of the values and equations seem to be arbitrary: Where does equation (2) come from? Are the values for K_dSM of 10 and 30 pu based on trial and error?**

The equation (2) in the paper is the standard equation of the power coefficient of a wind turbine for more details, refer [4]-[5]. Reference [5] is cited in line 71 as the basis for the derivation of the power coefficient, cp.

Yes, the value of $K_{d_{SM}}$ is obtained by a trial-and-error method for a simplified second-order model of a synchronous machine.

The explanation given to the response of referee 1 comments is repeated here to explain the method for determining $K_{d_{SM}}$:

In the simplified model, $K_{d_{SM}}$ represents the effect of the dampers of the synchronous machine, whose effects are neglected in this simplified modeling. This is a typical modeling which is used when the study does not need detailed information of the internal behaviour of the synchronous machine

Therefore, $K_{d_{SM}}$ is extracted from a detailed linearized system model through eigenvalue analysis. The damping coefficient obtained from this linear analysis is then validated with nonlinear time-domain simulations. This approach allows the extraction of an equivalent damping coefficient $K_{d_{SM}}$ associated with the synchronous machine modes [2].

To explain this we have modified sentence on line 209 as following:

**"** The damping coefficient $K_{d_{SM}}$ represents the combined damping effects in the system, which include mainly the effect of the synchronous machine's damper windings, neglected in this simplified model."

**3) What is an AC event (abstract)?**

In the context of this study, an "AC event" refers to a disturbance or dynamic condition occurring on the AC side of the power system, such as:

- Voltage dips or sags due to faults,
- Sudden changes in load or generation,
- Frequency deviations,
- Grid faults (e.g., line-to-ground or three-phase faults),
- Switching transients.

These events can lead to electromechanical oscillations, affect the inertial response of grid-forming converters, and influence the dynamic coupling between wind turbines and synchronous machines. The strategies proposed in the paper aim to mitigate such effects and improve the system's dynamic performance during these AC-side disturbances.

The line 4 is modified to include the AC event context as follow:

"This study explores the nonlinear coupling between wind turbines and AC grids and propose strategies for the enhancement of the inertial effect and the mitigation of oscillations which can arise in case of an **AC event-such as a grid fault or sudden load change.**"

**4) Could delta omega in equation (6) be defined?**

Thank you for your observation. In Equation (6), $\Delta\omega$ is an intermediate variable that represents the relative speed deviation between the turbine and generator sides of the wind turbine shaft.

Specifically, it is defined as:

$$\Delta\omega = \Omega_T - \Omega_G$$

This quantity reflects the torsional dynamics of the shaft in the two-mass model of the wind turbine system, as illustrated in Figure 2. Following sentence is included in the manuscript in Section 2.1.1, line 88

" ..., and $\Delta\omega$ represents the difference in rotational speed between the turbine $\Omega_T$ and the generator $\Omega_G$."

5) **Is the wind speed for all wind turbines assumed to be constant? I would assume so, because besides the load change, everything seems to be constant.**

Yes, the wind speed for all wind turbines is assumed to be constant in this study.

We choose to keep the wind speed constant in order to analyze the system dynamics more effectively. Varying the wind speed would change both the system dynamics and the operating point, making the analysis more complex. Therefore, to conduct a thorough analysis, we focus on modifying the grid side while maintaining a constant wind speed and hence a steady operating point.

To include this, sentence on line 71 is modified as follows:

"The wind farm consists of 180 type IV wind turbines of 5MW, **with constant wind speed at each turbine**"

6) **Is it correct that in the end the only effect of the load change of 3% or 20% is that the synchronous generator in the conventional power station increases its load?**

Yes, that is correct. In the system setup considered, the governor of the synchronous machine includes a frequency droop control which means that the active power of the synchronous machine will be adjusted with respect to the frequency.

For the converter, there not droop control, hence, in case of a frequency variation, the GFM converter provides an instantaneous inertial effect but it comes back to the original set point when the frequency is stabilized. Hence, it means that the synchronous machine compensate alone, in steady state, the full power difference of the load.

It could be possible to add a frequency droop control on the GFM but it is out of the scope of the paper.

This behaviour is reflected in the simulation results shown in Figure 4 and Figure 5 of this document, where the synchronous generator's active power increases in response to the step change in load.

7) **In the graphs in figure 7 and 8, I see that the frequency decreases, but I do not see that the power generated by the synchronous generator increases. Why (not)?**

In fact Fig 7. And 8 and fully dedicated to the wind turbine

$\Delta P_{AC}$: The AC power variation for the wind turbine

$\Delta P_G^*$: The power references for the wind turbine Generator

$\Delta P_T$ : The mechanical power of the wind turbine

As it has already been explained, only the synchronous machine modifies its operating point to compensate the load variation.  It is the reason why all these 3 power variations come back to 0 after the transient.

8) **The paper suggests some solutions for the problems. Are these the only possible solutions, the best solutions, or just solutions that (accidentally) happen to work?**

   The solutions proposed in the paper are not claimed to be the only or globally optimal solutions, but rather practical and effective strategies that have been derived from the model..

   As explained in comment 1, a model has been developed which highlights the coupling between the wind turbine level and the power system level. If the wind turbine is required to provide inertial effect which is beneficial for the grid, the energy which is provided to the grid in case of a frequency variation is provided by the mechanical part of the wind turbine. But the MPPT will try to counteract this effect and then decrease the inertial effect provided to the grid. Adding a low pass filter (lag filter) in the loop allows to decouple both effects but induces in the same time a risk of instability.  Adding a "lead filter" brings stability but may also counteract the "lag filter" effect.  This is a trade off between several contradictory requirements.  Furthermore, the non-linear nature of the system adds complexity to the approach, requiring verification that the tuning is effective at various operating points.

   The design approach is based on a combination of analytical insight from a simplified linear model and validation through nonlinear time-domain simulations. While other solutions may exist, the methods presented here are intended to demonstrate a systematic approach to identifying and mitigating specific dynamic coupling issues in such systems. We recognize that there may be other solutions with advanced filters, and exploring these would be a valuable extension of this work.

9) **In parts of Europe, there are times that significantly more than half of the generation comes from Inverter Based Resources (IRBs). Therefore, it seems that this research is describing things that are already used in the industry. Is that correct? Does this mean that science is lagging with**

**the application? Or do we apply things we do not yet sufficiently understand?**

Thank you for raising this important point. It is true that in parts of Europe and other regions, inverter-based resources (IBRs) — particularly wind and solar — already contribute significantly to power generation. However, many of the grid-forming capabilities required for stable operation under high IBR penetration are still emerging, and their widespread deployment remains limited.

While industry is actively deploying IBRs, the majority still operate in grid-following mode, relying on the presence of synchronous machines to set system frequency and voltage. Grid-forming control, which enables IBRs to contribute to system strength, inertia, and stability, is only recently being piloted and is not yet fully standardized or broadly implemented. As already mentioned before, this work is closely linked with the new requirements, under discussion between TSOs and manufacturers, that will be included in the new grid codes.

Therefore, this research addresses critical gaps in our understanding of how IBRs — especially grid-forming converters — interact dynamically with synchronous machines and wind turbines. In this sense, science is not lagging, but is instead working in parallel with industry to provide the modelling, analysis, and control design tools necessary to support a secure transition toward IBR-dominated systems.

We believe our study contributes to this effort by offering analytical insight and control strategies that can inform both current and future grid-forming applications.

10) **Do the problems (the oscillatory behavior) described in the paper also show up in practice? If yes, comparison with experimental results would be a valuable form of experimental validation of the work. If no, what does that say about this paper?**

Thank you for raising this important point. It is true that inverter-based resources (IBRs) — especially wind power plants (WPPs) — contribute substantially to power generation in various regions. However, many grid-forming capabilities essential for stable operation at high IBR penetration remain under active development and are not yet widely deployed.

Currently, most WPPs operate using grid-following control with phase-locked loops (PLLs). This control approach has been associated with torsional oscillations and subsynchronous oscillations (SSOs), particularly in weak grid conditions. The West China case notably reported 30-Hz oscillations occurring at only 5% of WPP capacity, which were linked to poor PLL tuning . Following this, [6] analysed the underlying mechanisms through electromagnetic transient simulations and confirmed that inappropriate PLL parameters can excite high-frequency SSOs that interact with the torsional modes of thermal generators. This interaction can exacerbate mechanical stress and stability concerns in the power system.

These findings highlight the critical challenge posed by torsional oscillations induced or amplified by grid-following converters in weak grids. In contrast, grid-forming converters—which provide direct control over voltage and frequency references without relying on PLL synchronization—offer promising pathways to mitigate such oscillations.

Our research aims to bridge these gaps by investigating the dynamic interplay between grid-forming converters, synchronous machines, and wind turbines, with a focus on mitigating oscillatory behaviours including torsional interactions. By providing analytical insights and control strategies, we support the ongoing transition toward more stable, inverter-based power systems.

Even if there is no experimental validation, we believe this work is a valuable step toward ensuring secure and reliable operation as power grids evolve toward higher penetration of inverter-based generation.

11) **If we want to go through energy transition, the challenge is not to keep a stable power system while there is always enough reserve power form thermal power stations, but to keep a stable power system fully based on renewable energy (IRBs) with all the intermittency, so that is where I see the real challenges.**

We fully agree with the reviewer that the long-term challenge of the energy transition lies in achieving a stable and resilient power system based entirely on renewable energy and inverter-based resources (IBRs) — especially given their inherent variability, limited inertia, and complex control dynamics.

Our study addresses an intermediate but crucial step in this transition: understanding the dynamic interactions between grid-forming converters and synchronous machines, and identifying control strategies to mitigate oscillations and enhance system support under disturbances. These insights remain highly relevant in the near-to-medium term, as hybrid systems (with both synchronous and inverter-based generation) are expected to dominate during the transitional phase. As already mentioned , this is a real concern for the TSOs and it is the reason why the grid codes will be modified in the coming years to introduce grid forming in the

different devices which are connected to the grid. Grid forming for batteries is not a huge issue and this is already in operation but grid forming for wind turbine is much more challenging due to the strong consequences on the mechanical part of the wind turbine.

That said, we recognize that future research must extend beyond this hybrid context to fully IBR-based systems. We consider our work as a foundation that can be built upon to address more complex scenarios involving multiple grid-forming units, distributed control, and renewable intermittency.

**Additional References**

[1] ENTSOE "Frequency Stability Evaluation Criteria for the Synchronous Zone of Continental Europe", March 2016

[2] P. Kundur, "Power System Stability and Control," McGraw- Hill, New York, 1994.

[3] Rokrok, E.: Grid-forming control strategies of power electronic converters in transmission grids : application to HVDC link, PhD, Centrale 365 Lille Institut, https://theses.hal.science/tel-04041405, 2022

[4] Heier, S. (2014). Wind Energy Conversion Systems. In Grid Integration of Wind Energy, S. Heier (Ed.). https://doi.org/10.1002/9781118703274.ch2

[5] Artur Avazov. AC Connection of Wind Farms to Transmission System : from Grid-Following to Grid-Forming. Electric power. Centrale Lille Institut; Katholieke universiteit te Leuven (1970-..). Faculteit Geneeskunde, 2022. English. ⟨NNT : 2022CLIL0024⟩. ⟨tel-04074095⟩

[6] Y. Li, L. Fan and Z. Miao, "Wind in Weak Grids: Low-Frequency Oscillations, Subsynchronous Oscillations, and Torsional Interactions," in IEEE Transactions on Power Systems, vol. 35, no. 1, pp. 109-118, Jan. 2020, doi: 10.1109/TPWRS.2019.2924412. keywords: {Phase locked loops;Oscillators;Mathematical model;Power system dynamics;Analytical models;Wind farms;Voltage control;Type-4 wind;phase-locked loop (PLL);torsional interaction;low-frequency oscillations;subsynchronous oscillations},